# Molecular mechanisms of coronary artery disease risk at the *PDGFD* locus

Hyun-Jung Kim [1], Paul Cheng [1], Stanislao Travisano [1], Chad Weldy[1], João P. Monteiro [1], Ramendra Kundu[1], Trieu Nguyen [1], Disha Sharma[1], Huitong Shi[1], Yi Lin[2,3,4], Boxiang Liu [3,4], Saptarsi Haldar[5], Simon Jackson[5] & Thomas Quertermous [1] ✉

Genome wide association studies for coronary artery disease (CAD) have identified a risk locus at 11q22.3. Here, we verify with mechanistic studies that rs2019090 and *PDGFD* represent the functional variant and gene at this locus. Further, FOXC1/C2 transcription factor binding at rs2019090 is shown to promote *PDGFD* transcription through the CAD promoting allele. With single cell transcriptomic and histology studies with *Pdgfd* knockdown in an SMC lineage tracing male atherosclerosis mouse model we find that Pdgfd promotes expansion, migration, and transition of SMC lineage cells to the chondromyocyte phenotype. Pdgfd also increases adventitial fibroblast and pericyte expression of chemokines and leukocyte adhesion molecules, which is linked to plaque macrophage recruitment. Despite these changes there is no effect of *Pdgfd* deletion on overall plaque burden. These findings suggest that *PDGFD* mediates CAD risk by promoting deleterious phenotypic changes in SMC, along with an inflammatory response that is primarily focused in the adventitia.

Coronary artery disease (CAD) is predicted to continue as the worldwide leading cause of human mortality for at least the next two decades[1,2]. While as much as half of the disease risk is conferred by classical risk factors that have been ameliorated by the development of targeted therapies, the remainder of the risk is still unaddressed. Genome wide association studies (GWAS) have identified hundreds of genomic loci that contribute to the genetic risk for CAD, with further studies indicating that genes in these loci regulate the primary cellular processes that underlie the remaining disease risk through their effect on vascular wall cellular and molecular mechanisms, as well as disease related processes in liver and adipose tissues[3–6]. These data suggest that investigation of the molecular pathways that are embedded in CAD gene regulatory networks will provide new and effective approaches to treating this devastating disease. Indeed, there are currently no drugs that effectively target the primary disease process in the vessel wall.

Recent GWAS meta-analyses have identified approximately 250 loci that confer CAD risk[7,8]. While only a handful of these loci have been studied thus far, it is increasingly clear that smooth muscle cells (SMC), endothelial cells and macrophages confer a significant portion of the genetic disease risk[9], through phenotypic transitions that are mediated by dramatic cell state changes[10–14]. For SMC, these phenotypic changes have been linked to disease risk through single cell RNA sequencing (scRNAseq) and cellular lesion anatomy studies showing that expression of protective CAD associated gene *Tcf21* promotes transition primarily to the fibroblast-like fibromyocyte (FMC) phenotype, and that protective Tgfb signaling molecules Zeb2 and Smad3 fundamentally alter or inhibit transition to the chondrocyte-like chondromyocyte (CMC) phenotype[12,15,16]. While atherosclerosis has been characterized as a primarily inflammatory disease[17], there has been a dearth of such molecules linked to the disease process by human GWAS studies.

[1]Division of Cardiovascular Medicine, 300 Pasteur Drive, Falk CVRC, Stanford, CA 94305, USA. [2]Research Center for Intelligent Computing Platforms, Zhejiang Laboratory, Hangzhou 311121, China. [3]Department of Pharmacy, Faculty of Science, National University of Singapore, Singapore 117543, Singapore. [4]Department of Biomedical Informatics, Yong Loo Lin School of Medicine, National University of Singapore, Singapore 119228, Singapore. [5]Amgen Inc., 1120 Veterans Blvd, South San Francisco, CA 94080, USA. ✉e-mail: tomq1@stanford.edu

Although not guided by human genetic data, platelet-derived growth factors (PDGFs) have been implicated in the fundamental biology of vascular wall development as well as the pathophysiology of atherosclerosis[18,19]. PDGFs were originally identified in platelets and serum as potent mitogens for smooth muscle cells and fibroblasts in vitro[20,21]. The PDGF family consists of four ligands, A–D, forming dimeric proteins that signal through two tyrosine kinase receptors, PDGFRA and PDGFRB. The ligands and receptors can form homodimers or heterodimers depending on cell type, receptor expression, and ligand availability[22–25]. Interestingly, the most recently characterized ligand PDGFD can bind PDGFRB homodimers, PDGFRA-PDGFRB heterodimers as well as heterodimers involving NRP1 and PDGFRB[25]. Signaling through PDGFRB has been shown to initiate endothelial, pericyte, and smooth muscle cell proliferation and migration both in vitro and in vivo[23,24]. The PDGFB and PDGFRB system is critical for the migration and proliferation of pericytes and the development of a functional vasculature[26,27]. Deletion of *Pdgfrb* in the disease setting has been shown to abrogate the SMC cell state changes that represent the response of this cell type to disease stimuli[28].

The locus encoding *PDGFD* has been identified in GWAS studies to be associated with CAD risk[8,29]. However, biological investigation of a role for *PDGFD* in atherosclerosis has yet to be conducted. Here, through fine mapping approaches we present data suggesting that *PDGFD* is the disease gene for CAD at this locus and further provide evidence to support the mechanism of association to be due to FOXC1/C2 differential binding at the rs2019090 associated variant. By generating a *Pdgfd*[-/-] mouse model on an atherosclerosis genetic background with SMC lineage tracing combined, single cell transcriptomics and lesion anatomy studies, we show that this factor modulates SMC expansion, phenotypic transition, and migration into the plaque with additional effects on monocyte recruitment and vascular inflammation. Together, we provide evidence that supports *PDGFD* as the disease gene at this CAD risk locus and reveal insights into its role in mediating vascular smooth muscle specific phenotypic changes and plaque biology.

## Results

### Fine mapping and epigenome editing at the 11q22.3 CAD GWAS locus implicates rs2019090 as the functional associated variant and *PDGFD* as a disease gene

Our group previously identified 87 candidate genetic variants that are associated with CAD, using human coronary artery smooth muscle cell (HCASMC) ATAC-seq and ChIP-seq data with CARDIoGRAMplusC4D CAD GWAS variants[30]. After filtering for variants with a combination of known and predicted regulatory elements in the intergenic regions and evidence of transcription factor binding in vivo, we prioritized 64 variants in HCASMC. Six CAD SNPs in high linkage disequilibrium were noted to be associated to CAD risk by GWAS at 11q23.2. One of these SNPs, rs2019090, was localized 150 kilobases (kb) downstream of *PDGFD* in an intron of the long non-coding RNA (lncRNA) *AP002989.1*, in HCASMC peaks for ATAC-seq identified open chromatin and enhancer related H3K27ac histone modification, in juxtaposition to ChIP-seq peaks for CAD transcription factors (TFs) TCF21 and SMAD3 (Fig. 1a–c). GWAS data curated in the NHGRI-EBI GWAS Catalog (V1.0.2) also indicated association with carotid intimal-medial thickness (IMT), with the A allele identified as promoting disease risk for both CAD and IMT[31–34]. Rs2019090 was shown to serve as an expression quantitative trait locus (eQTL) variant in analysis of GTEx data ($p = 1.6e-8$), with the risk 'A' allele being associated with greater expression of *PDGFD*[5,32,35,36], as well as increased expression of *AP002989.1* in an early GTEx analysis of aortic tissue ($p = 3.72e-5$)[37]. In addition, this variant was identified with GTEx data as a splicing QTL (sQTL) for *AP002989.1* ($p = 4.1e-8$). Further, mapping of recent CAD GWAS association findings to vascular eQTL data using the locuscompare.com visualization tool (locuscompare.com) suggested that rs2019090 provides the greatest

contribution to CAD risk and *PDGFD* expression (Fig. 1d and Supplementary Fig. 1a). This was validated with the *enloc* fine mapping and genome-wide co-localization analysis algorithm[38]. This tool implements a Bayesian hierarchical model with summary level data to provide a rigorous statistical evaluation of related genome-wide traits, and for these studies identified a significant relationship between CAD GWAS meta-analysis summary level data and GTEx vascular tissue eQTL data (Supplementary Fig. 1b). *PDGFD* was identified as significant with a regional level colocalization probability (RCP) of 0.2 as the recommended cutoff to select significant colocalization between the GWAS and eQTL data[39].

Further, we have examined the DNA sequence at rs2019090 and found that this SNP is localized in a putative FOXC1/C2 binding site. Searches of relevant TF position weight matrices (PWMs) in the JASPAR database[40] with the motif comparison MEME Suite tool Tomtom[41] have found a significant match ($p = 3e-3$) for the two highly homologous TFs FOXC1 and FOXC2 (Fig. 1c, e). Interestingly, the rs2019090 polymorphism is multi-allelic. A is the reference allele but *T* is the alternate allele in European cohorts with C and G serving as additional alternatives. As evident from the FOXC1/C2 PWMs, both A and G are common at the SNP site, and *T* is the least common base, suggesting that replacement of A with T by the rs2019090 variant would decrease FOXC1/C2 binding and expression of the target *PDGFD* gene (Fig. 1e). Both *FOXC1* and *FOXC2* reside in loci found to be associated with CAD[7,8], although definitive experiments have not been conducted to prove that they are the disease associated genes in their respective loci.

To experimentally investigate whether *PDGFD* is the disease-related gene at 11q22.3, we employed epigenetic targeting at the rs2019090 variant. CRISPRi was conducted by transducing HCASMC with the AA genotype, lot 2897, with lentiviruses encoding dCas9KRAB along with one of three single guide RNAs (Supplementary Fig. 1c). Gene expression was evaluated by quantitative real-time PCR, for both *PDGFD* and the lncRNA *AP002989.1* (Fig. 1f, g). These experiments indicated that *PDGFD* expression was highly significantly suppressed by two of the three guides, but interestingly the lncRNA expression was not affected. It is a consideration that CRISPRi with this approach suppresses over a distance of 1–2 kilobases, but there are no other protein coding genes within 100,000 base pairs of the targeted region. These findings support the identification of *PDGFD* as the disease-associated gene and indicate that lncRNA *AP002989.1* is not a direct target of the disease association mechanism.

### FOXC1 regulates *PDGFD* expression via functional CAD-associated SNP rs2019090 to establish a complex gene regulatory network

We evaluated allele-specific transcription of the rs2019090 enhancer region by FOXC1 and FOXC2, with dual luciferase assays in the A7r5 rat vascular smooth muscle cell line. Three copies of the 150 bp region of the *AP002989.1* intron flanking the candidate SNP rs2019090, containing either the A or the T allele, were cloned into a luciferase reporter construct and co-transfected with *FOXC1* or *FOXC2* expression constructs. These and other in vitro assays were performed at least three times with each having at least three biological replicates. Luciferase activity showed that over-expression of *FOXC1* and *FOXC2* significantly activated the A allele but suppressed the T allele reporter, indicating a direct and allele-specific regulation by FOXC1 and FOXC2 (Fig. 2a, b). While both of these TFs reside in CAD associated loci, and thus may be directly linked to CAD risk[8], we have decided to focus subsequent studies on *FOXC1*. The transcriptionally active A allele is more highly represented in its binding sites, *FOXC1* mutations have been linked to PDGF signaling in the context of cerebral small vessel disease[42], and this gene has also been linked to vascular risk factors including hypertension, systolic blood pressure, and waist hip ratio (GWAS catalog). *FOXC2* has been related primarily to non-vascular

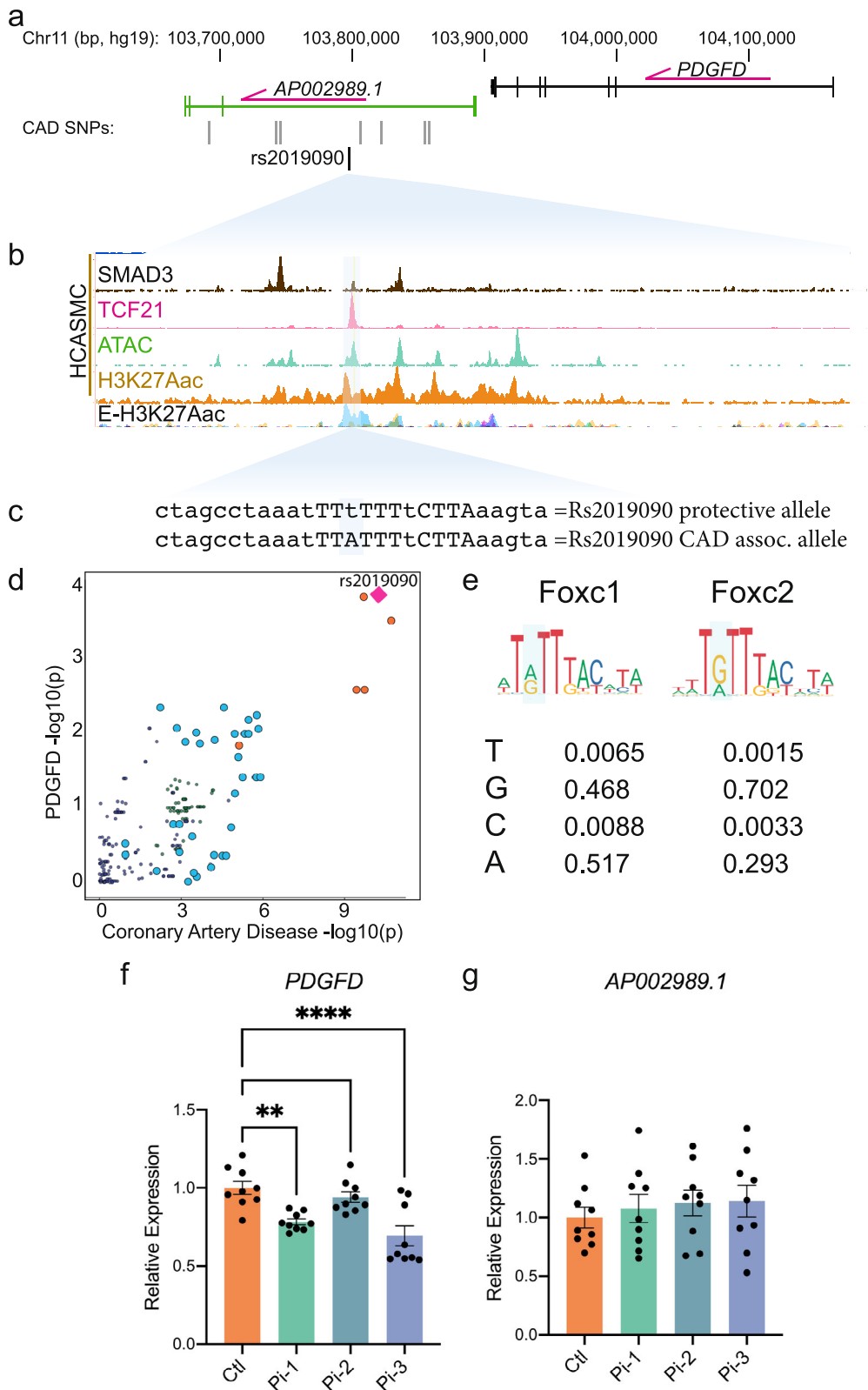

phenotypes, including cortical thickness and white matter hyper-intensity volume (GWAS catalog).

To determine *FOXC1* allele-specific cis-effects on endogenous *PDGFD* expression, we performed short interfering RNA (siRNA)-mediated knockdown (KD) or lentivirus-mediated overexpression (OE) of *FOXC1* in four different primary HCASMC lots known to have AA, AT, or TT genotypes at rs2019090 (Fig. 2c)[43]. We found that *PDGFD*

expression is decreased with *FOXC1*-KD and increased with *FOXC1*-OE in both A/A homozygous and A/T heterozygous but not in T/T homozygous HCASMC, indicating that endogenous FOXC1 positively regulates *PDGFD* expression through the A allele of SNP rs2019090. Overall, these results were consistent with the enhancer trap assays and suggested that FOXC1 promotes *PDGFD* expression through the disease-associated A allele. With the knockdown studies we did not

**Fig. 1 | Functional mapping of candidate 11q22.3 locus proposes regulatory mechanisms of PDGFD expression and disease association. a** UCSC browser screenshot at 11q22.3 locus showing position of *PDGFD* gene and lncRNA *AP002989.1* relative to the candidate SNP rs2019090, and **b** overlap of rs2019090 with ChIP-seq tracks for CAD risk transcription factors SMAD3 and TCF21. Also shown are ATAC-seq open chromatin and active enhancer histone modification H3K27ac ChIP-seq tracks in human coronary artery smooth muscle cells (HCASMC), as well as ENCODE layered H3K27ac for HUVEC (blue) and NHLF (purple) cells. Genomic coordinates refer to hg19 assembly. **c** Genomic sequence at rs2019090 for protective and disease alleles, with FOXC1/C2 motifs indicated. **d** Co-localization of coronary artery disease (CAD) GWAS association data on the *x*-axis and *PDGFD* eQTL association data (GTEx v8, aorta) on the *y*-axis. *P*-values represent original GWAS findings, not a statistical test to determine variant causality. **e** Position weight matrices for FOXC1 and FOXC2, as per the JASPAR database. **f, g** CRISPRi epigenetic silencing by transduction of dCas9KRAB and single guide RNAs targeted around rs2019090 in cell lot 59386145, a HCASMC primary culture with AA genotype. Expression of *PDGFD*, ** $p = 0.0033$, **** $p < 0.0001$ and lncRNA *AP002989.1* were evaluated by quantitative RT-PCR. Data were normalized relative to control and expressed as mean ± s.e.m. with *p*-values obtained using ordinary one-way ANOVA with Dunnett's multiple comparisons post-hoc test. Dots represent three technical replicates from three biologically independent samples. Source data are provided in the Source Data File.

find evidence that FOXC1 suppresses *PDGFD* expression through the T allele at SNP rs2019090 in T/T homozygous HCASMC. Given that rs2019090 is located within the structural *AP002989.1* lncRNA, and eQTL studies have associated splicing of this lncRNA with genotype at rs2019090, we performed similar studies examining the effects of *FOXC1* perturbation on the expression of this gene. In contrast to the results for *PDGFD* expression, neither increased or decreased *FOXC1* expression altered mRNA levels for *AP002989.1* (Fig. 2d).

We further investigated the regulatory relationship among members of the FOXC1-PDGFD pathway. We found that *FOXC1* expression is significantly increased with *PDGFD*-KD (Fig. 2e, f) suggesting a negative regulatory interaction between these factors and supporting their pathway relationship. *PDGFD*-KD in HCASMC did not show a significant change in the expression of *AP002989.1* (Fig. 2g). Receptors are commonly counter-regulated by ligand levels and we investigated the expression of the two receptors known to bind PDGFD. Both the *PDGFRA* and *PDGFRB* receptor genes showed upregulation with knockdown of *PDGFD* (Fig. 2h, i), further linking these factors in a functional PDGFD signaling pathway in SMC. To complement these loss of function studies in HCASMC, we performed gain of function studies by lentivirus transduction to over-express *PDGFD* in these cells. We grouped batches of HCASMC expressing varying levels of PDGFD after transduction with lentivirus, dividing them into tertiles for low, moderate and high expression levels, and used quantitative RT-PCR to study the transcriptional response of related factors to increased PDGFD. We identified and employed viral titers that provided low-, medium-, and high-level expression of *PDGFD* (Fig. 2j). In keeping with interactions identified with *PDGFD*-KD, expression levels of *FOXC1*, *PDGFRA* and *PDGFRB* genes showed the opposite response to PDGFD by decreasing their expression (Fig. 2k, m, n). *AP002989.1* expression level was reduced, though not significantly, in response to increased *PDGFD* expression (Fig. 2l).

## *Pdgfd* promotes SMC phenotypic transitions as well as monocyte-macrophage recruitment

To investigate the cellular and molecular mechanisms by which PDGFD regulates atherosclerosis development and CAD risk we developed a mouse atherosclerosis model that provided constitutive knockout of *Pdgfd*, as well as SMC-specific lineage marking in the *ApoE*$^{-/-}$, *C57BL/6* background. A constitutive *Pdgfd* knockout (KO) mouse allele that was previously generated by replacing exon1 in the *Pdgfd* gene with a *LacZ* expression cassette[44] was combined with a Cre-activatable tandem dimer Tomato (*tdT*) fluorescent marker gene in the *ROSA26* locus[45,46], and the highly SMC-specific *Myh11*-Cre recombinase transgenic allele[12,47,48], in the atherogenic *ApoE* knockout (*Pdgfd*$^{lacZ/lacZ}$, *Myh11*$^{CreERT2}$, *ROSA*$^{floxtdT/+}$, *ApoE*$^{-/-}$, designated KO). Lineage tracing allows for highly efficient and permanent labeling of smooth muscle cells, and their progeny, with *tdT* during subsequent cell state changes[12,15,16]. Both *Pdgfd* KO and control (*Pdgfd*$^{+/+}$, *Myh11*$^{CreERT2}$, *ROSA*$^{tdT/+}$, *ApoE*$^{-/-}$, designated as Ctl) mice were administrated tamoxifen at the age of 8 weeks, followed by high-fat diet (HFD) feeding for 16 weeks to induce atherosclerosis (Fig. 3a). We did not observe significant differences in total body weight with HFD

feeding compared to wild-type control mice as reported previously[44]. Atherosclerotic lesions in the aortic roots were dissected, tissue digested, and isolated cells subjected to FACS (Supplementary Fig. 2a) to separate aortic tdT positive and negative cells from both KO (three groups, two mice each group) and Ctl mice (two groups, two mice each group), employing methods that we have previously described[12]. Cells were captured with the 10× Genomics Chromium microfluidics device and libraries generated and sequenced as described (Fig. 3a)[12].

After quality control assessment, scRNAseq data were visualized using uniform manifold approximation and projection (UMAP) dimensionality reduction plots (Fig. 3b). Unsupervised clustering analysis at the optimal 2.6 resolution parameter identified a total of 13 clusters and cell-specific markers used to identify cluster lineages (Fig. 3b, Supplementary Fig. 2b, c, and Supplementary Data 1). Lineage traced cells were identified by *tdT* expression and noted to contribute to four separate clusters as we have described previously: SMC, fibroblast-like fibromyocytes (FMC), endochondral bone like chondromyocytes (CMC), as well as pericytes[15,16]. Quiescent and transition SMC clusters were readily separated from other clusters with low resolution parameters. At this resolution, endothelial cells, and fibroblasts each contributed to two separate clusters, Endo-1 vs Endo-2 and Fblst-1 vs Fblst-2 respectively, as previously described[15,16]. Feature plots (Fig. 3c, d and Supplementary Fig. 2d) and violin plots (Supplementary Fig. 2e) were employed to visualize the cluster-specific expression of *Pdgfd* and *Pdgfb*, as well as *Pdgfra and Pdgfrb*. In lineage traced cells in control tissue, *Pdgfd* was expressed in SMC and FMC, but showed lower expression levels in CMC. In non-SMC lineage cells, there was robust expression in pericytes, Endo-1 and epithelial cells, and modest expression in fibroblasts. Interestingly, *Pdgfrb* was expressed in all SMC lineage cells, including the CMC, as well as Fblst-1, Fblst-2 and pericyte cluster cells. Knockout of *Pdgfd* produced an apparent increase in SMC and decrease in transition CMC (Fig. 3e), and also a modest decrease in *Pdgfrb* expression in all cells expressing significant levels of *Pdgfd* (Supplementary Fig. 2f). We also analyzed the average expression of *Pdgfra* as well as other PDGF ligands *Pdgfa* and *Pdgfb* but found no significant change in their expression level in vascular cells (Supplementary Fig. 2f).

To examine changes in relative cluster cell numbers, we measured the average percentage of cells in different clusters separately for lineage traced and non-lineage traced cells (Fig. 3f, g). In traced cells, loss of *Pdgfd* increased the relative proportion of differentiated SMC but decreased FMC and CMC cluster numbers (Fig. 3f). Importantly, in non-lineage cells, loss of *Pdgfd* resulted in a decrease in macrophage number (Fig. 3g). While this analysis indicated a relative increase in fibroblasts among non-*tdT* lineage traced cells, the absolute number was the same for both genotypes (0.47% versus 0.51%). Also, it is important to note that the relative representation of adventitial cells in scRNAseq experiments is highly variable due to differences in extent of adventitial tissue included in the aortic tissue isolation. Together, these data indicate that loss of *Pdgfd* inhibits SMC phenotypic transition and monocyte-macrophage recruitment during atherosclerosis development.

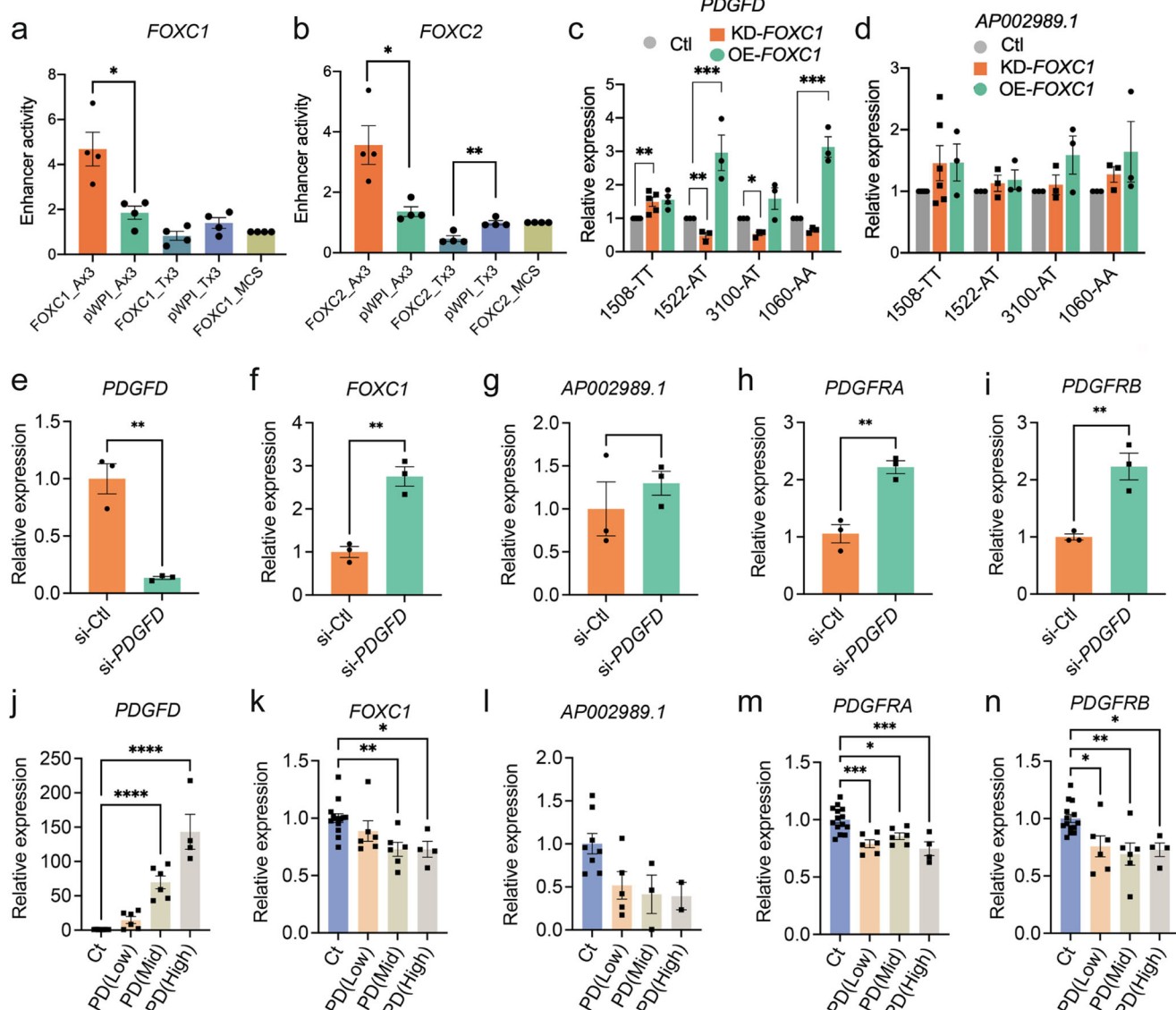

**Fig. 2 | FOXC1 regulates PDGFD expression via functional SNP rs2019090 to establish a complex gene regulatory network.** Results of enhancer trap assay for **a** *FOXC1*, * *p* = 0.0126 and **b** *FOXC2*, * *p* = 0.0155, ** *p* = 0.0040, or empty control (pWPI) co-transfected with luciferase reporters with three copies of the 150 base pair region containing the A allele (Ax3) or T allele (Tx3) cloned into the minimal promoter-driven luciferase reporter vector pLUC-MCS (MCS). A7r5 rat vascular smooth muscle cells were used for these assays. Values represent mean ± s.e.m. of four biologically independent experiments expressed as fold change relative to pWPI-empty plasmid with p-values obtained with two-sided unpaired t-test. **c** Results of quantitative polymerase chain reaction (qPCR) analysis for *PDGFD*, *n* = 5 biologically independent knockdown samples, ** *p* = 0.0026, ** *p* = 0.0058, * *p* = 0.0123, and *n* = 4 biologically independent over-expression samples in 1508 cell lot and *n* = 3 for all other lots and conditions; *** *p* = 0.0004, *** *p* = 0.0002. Analysis was performed using one-way ANOVA with Dunnett's multiple comparisons post-hoc test or **d** *AP002989.1* expression with knockdown (KD), *n* = 6 biologically independent knockdown samples in 1508 cell lot, or over-expression (OE) of *FOXC1*, *n* = 3 biologically independent samples for all other conditions and cell lots, in HCASMC carrying different genotypes for rs2019090. Each dot represents a biologically independent sample. Data were normalized relative to controls and expressed as mean ± s.e.m with *p*-values using ordinary one-way ANOVA with Dunnett's multiple comparisons post hoc test. **e** qPCR analysis for expression levels of *PDGFD*, ** *p* = 0.0028, **f** *FOXC1*, ** *p* = 0.0024, **g** *AP002989.1*, **h** *PDGFRA*, ** *p* = 0.0039, and **i** *PDGFRB*, ** *p* = 0.0068 with *PDGFD* knockdown (KD) in HCASMC. Each dot represents a biologically independent sample, *n* = 3. Data were expressed as mean ± s.e.m with *p*-values using a two-sided unpaired t-test. **j** qPCR analysis for expression levels of *PDGFD*, *n* = 6 for PD(Low) and PD(Mid), *n* = 4 for PD(High*)*, *** *p* < 0.001, **k** *FOXC1*, *n* = 6 for PD(Low) and PD(Mid), *n* = 4 for PD(High), ** *p* = 0.0055, * *p* = 0.0177 **l** *AP002989.1*, *n* = 5 for PD(Low), *n* = 3 for PD(Mid), *n* = 2 for PD(High), **m** *PDGFRA*, *n* = 6 for PD(Low) and PD(High), *** *p* = 0.0006, * *p* = 0.0189, *** *p* = 0.0004 and **n** *PDGFRβ*, *n* = 6 for PD(Low) and PD(Mid), *n* = 4 for PD(High), * *p* = 0.0244, ** *p* = 0.0035, * *p* = 0.0305 with *PDGFD* overexpression (OE) in HCASMC. Data grouped based on expression levels of *PDGFD* and expressed as mean ± s.e.m of biologically independent samples with *p*-values obtained from one-way ANOVA with Dunnett's multiple comparisons post-hoc test. Each dot represents a biological replicate. Data represented as relative expression as control ratio (treatment of scramble siRNA (si-Ctl, KD control) or empty-pWPI (Ctl, OE control). Source data are provided in the Source Data File.

## Pdgfd activates a broad gene expression program to establish SMC transition phenotypes and promote inflammatory pathway activation

Using the FindMarker algorithm of *Seurat*, we analyzed the scRNAseq data to identify genes that are differentially regulated with *Pdgfd* loss in comparison to control. Using a cutoff value set to 0.05 for the false discovery rate (FDR) *q*-value, a total of 165 transcripts were identified (Fig. 4a). While 58 transcripts were upregulated, 107 transcripts were downregulated, across all cellular phenotypes (Supplementary Data 2). Interestingly, more than half of upregulated genes belonged to

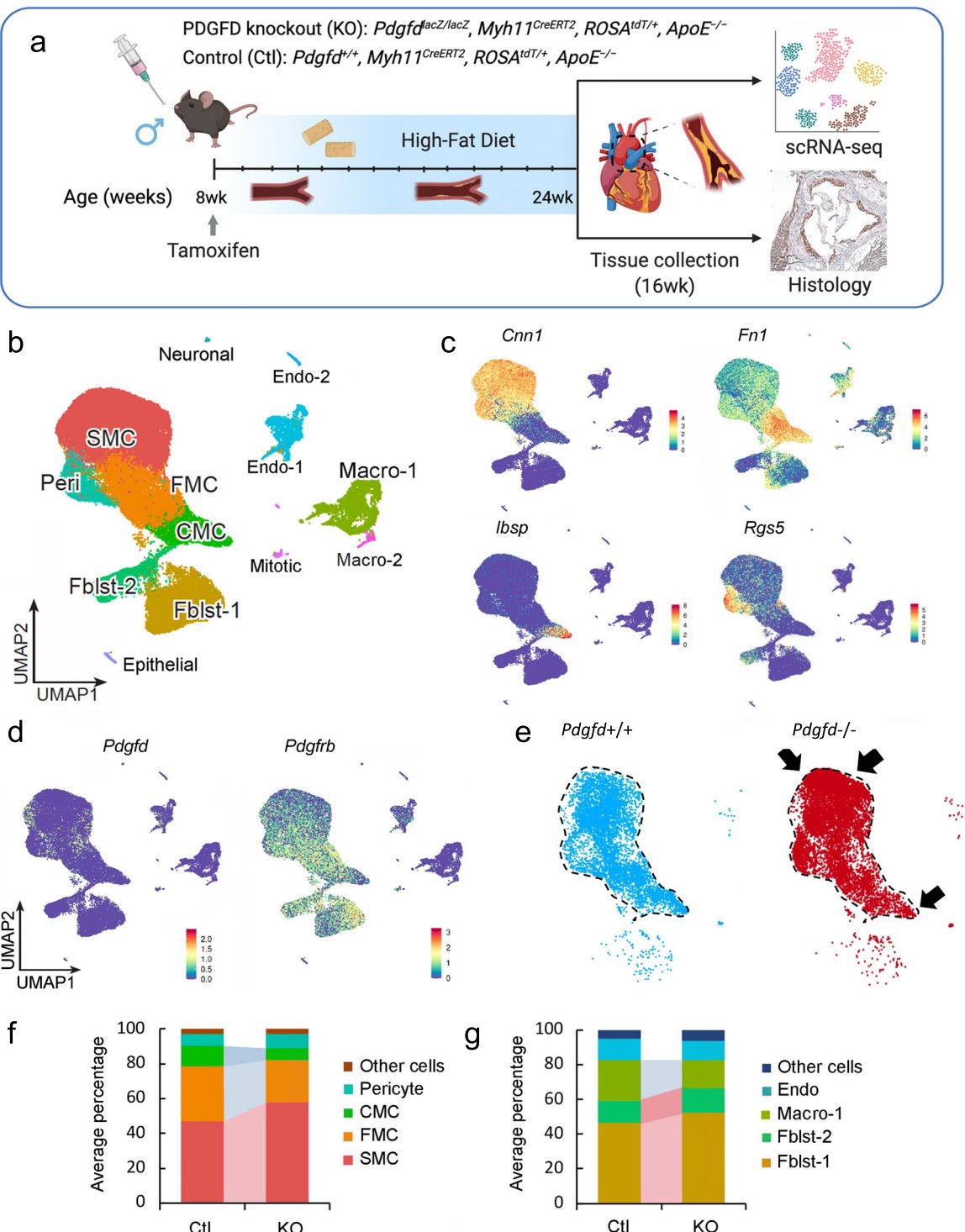

**Fig. 3 | Single-cell transcriptomic profiling of mouse atherosclerotic aortic root in Pdgfd KO mice. a** Schematic of experimental design showing that dissected aortic tissues were harvested for single cell RNA sequencing (scRNAseq) and histology analyses from SMC-specific lineage tracing control (Ctl) and lineage tracing *Pdgfd* knockout (KO) mice. Eight-week-old mice, 2 Ctl and 3 KO captures (two mice per capture), were treated with tamoxifen twice at 3-day intervals and subsequently fed high fat diet for 16 weeks and then sacrificed. Tissues were digested to single cells, tdTomato positive (tdT+) fluorescence and negative (tdT-) cells collected and captured on the10x Chromium controller, libraries generated and sequenced. Created with BioRender.com. **b** Uniform manifold approximation and projection (UMAP) of scRNAseq results identified 13 different clusters at 2.6 clustering resolution, with respective biological cluster identities as defined by cluster marker genes. **c** UMAP displaying expression of indicated markers reflecting unique cluster identity: *Cnn1*, SMC; *Fn1*, FMC; *Ibsp*, CMC; *Rgs5*, pericytes. **d** UMAP visualizing dimension reduction plots of *Pdgfd* and *Pdgfrb* expression. **e** UMAP images comparing feature expression of *tdTomato* positive cells from Ctl and KO mice. The dotted line is generated based on the Ctl image. Arrows indicate increase in SMC number and decrease in transition cell (CMC) number. **f** Bar plot presenting the average percentage of lineage traced cells and **g** non-lineage traced cells in Ctl and KO groups.

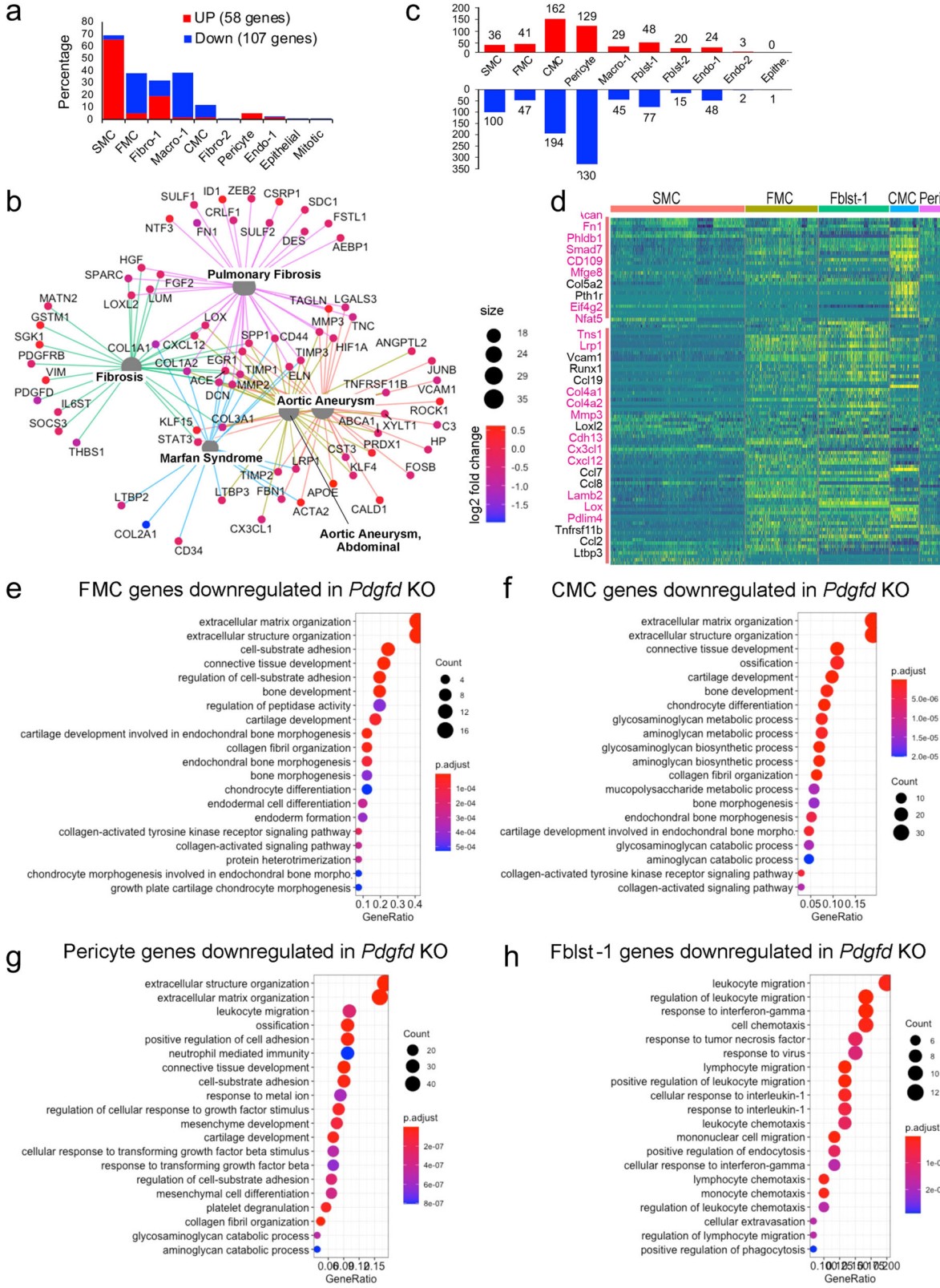

**Fig. 4 | Loss of Pdgfd mitigates the smooth muscle cell chondrogenic transition and inflammatory pathway activation. a** Bar plot showing the number of upregulated genes (58, red bars) and downregulated genes (107, blue bars) derived from all KO compared to all Ctl disease tissues. **b** Gene-disease network analysis of the differentially expressed genes (DEGs) among lineage traced cells in KO compared with Ctl as determined by gene set enrichment analysis with *clusterprofiler*, and shown as a gene-gene regulatory network with *enrichplot*. **c** Bar plot displaying numbers of DEGs in individual clusters, for KO compared with Ctl. **d** Heatmap showing expression patterns of downregulated DEGs across different cluster groups, based on fold-change of gene expression. Yellow color indicates differential expression, genes in red text reside in window of lead SNP ± 500 kilobases. **e–h** Graphs depicting gene set enrichment analysis underlying biological process of DEGs for **e** FMC, **f** CMC, **g** pericytes, and **h** CMCs as determined by *clusterProfiler*. Gene set enrichment analysis (GSEA) was used, with permutation test to determine adjusted p-value.

quiescent SMC (38/58, 65.5%), whereas most of the downregulated genes belonged to SMC-derived FMC (35/107, 32.7%) and CMC (11/107, 10.3%) as well as macrophage clusters (39/107, 36.4%). Two CMC markers, *Col2a1* and *Ibsp*, were the most overall highly decreased transcripts (Supplementary Data 2). We found an increase in SMC differentiation markers, such as *Acta2* and *Tagln*, but a decrease in FMC and CMC markers, such as *Vcam1* and *Col2a1*, respectively. These data suggest that *Pdgfd* enhances SMC de-differentiation and phenotypic transition as well as monocyte-macrophage recruitment in the disease setting. Using the Molecular Signatures Database (MSigDB) and gene set enrichment analysis, we predicted the Biological Pathways enriched with those up and downregulated transcripts (Supplementary Fig. 3a). This pathway analysis identified highly significant immune-related terms and cell-matrix interaction and vascular development terms as the top pathways associated with down- and upregulated genes, respectively. We further examined the potential functional implication of differentially expressed genes between KO and Ctl tissues for disease pathogenesis. Differentially regulated genes were enriched and highly interconnected in disease categories related to fibrosis and aortic aneurysm formation (Fig. 4b), suggesting that these pathways overlap those involved in *Pdgfd*-mediated atherosclerosis.

To examine how Pdgfd specifically affects gene expression in clusters of cells with similar phenotype, we dissected the number of differentially regulated genes in individual clusters comparing between KO and Ctl tissues (Fig. 4c, d and Supplementary Data 2). Enrichment of these genes in biological processes (BP) as annotated with gene ontology terms were identified separately for clusters of interest. SMC downregulated genes identified terms related to extracellular matrix assembly and organization and TGFB pathway signaling (Supplementary Fig. 3b). FMC downregulated genes were enriched for extracellular matrix terms, but importantly also genes related to early chondrogenic processes, as indicated by identification of terms bone development and cartilage development (Fig. 4e). Genes downregulated in these pathways included *Col1a1, Col2a1, Col5a1, Thbs1, Ccnd1,* and *Fbn1*.

Downregulated genes for the CMC transition phenotype were 4-fold greater in number than those identified for the FMC phenotype (Fig. 4c), and the differentially regulated genes assigned to pathways included *Acan, Col2a1, Col10a1, Sox6, Pth1r,* and *Scrg1*. The majority of GO BP terms enriched for CMC-regulated genes in the *Pdgfd* KO vascular tissue reflected a prominent role for this growth factor in transition of SMC to a chondrogenic phenotype, including ossification and chondrocyte differentiation (Fig. 4f). Terms for this chondrogenic transition phenotype showed greater gene ratios and lower p-values compared to FMC gene pathways. Also, the heatmap of differentially regulated genes per cluster showed a significant difference between FMC and CMC gene expression (Fig. 4d). Interestingly, downregulated genes for the CMC phenotype included a greater number of putative CAD GWAS genes compared to other clusters, including *Acan, Fn1, Mfge8, Phldb1, Smad7, Cd109, Eif4g2,* and *Nfat5*[3,7,8]. There were an equally large number of downregulated genes in the *Pdgfd* knockout compared to wildtype CMC clusters, but these genes were not enriched in pathways that were informative regarding Pdgfd function or disease mechanisms. Pathways identified included multicellular organism process and developmental process among other general terms.

Pericytes showed the greatest number of downregulated genes with *Pdgfd* deletion, and there was considerable overlap with SMC differentially expressed genes and pathways (Fig. 4g and Supplementary Data 2). Surprisingly, pericytes showed down-regulation of numerous genes also downregulated in FMC, including *Loxl2, Casp4, Col1a1,* and *Thbs1*, as well as several CMC genes, including *Timp3, Lox, Fgf2, Fgfr1, Col5a1, and Col5a2* (Fig. 4d). These differentially regulated genes contributed to enrichment of GO BP terms extracellular matrix

organization, positive regulation of cell adhesion, and ossification. There was also down-regulation of leukocyte recruitment and adhesion molecules such as *Cxcl1, Cxcl5, Cxcl12, Ccl19,* and *Alcam*, providing enrichment for terms leukocyte migration and neutrophil-mediated immunity. Further, pericytes with deletion of *Pdgfd* showed a decrease in *Tgfb2* expression, as well as genes related to Tgfb signaling, identifying terms cellular response to Tgfb and response to Tgfb. These data suggest that Pdgfd activates gene expression patterns in pericytes related to Tgfb signaling and those identified with SMC transition phenotypes.

Analysis of gene expression changes in the Fblst-1 cluster indicated that Pdgfd promotes a highly specific phenotypic program in these cells related to inflammatory cell recruitment (Fig. 4h and Supplementary Data 2). *Pdgfd* deletion was associated with downregulation of a broad range of inflammatory mediators that interact with monocytes, neutrophils, T and B cells, including chemokines *Ccl2, Ccl7, Ccl8, Ccl19, Cxcl12, Cxcl14, Cxcl16*, proinflammatory cytokine *Il6*, and acute phase reactants *C1s1, C3, and C4b*. Interestingly, fibroblasts upregulated genes in *Pdgfd* KO cells related to SMC phenotype, including *Acta2, Tagln*, and *Myl9*, suggesting that Pdgfd may inhibit the transition of fibroblasts to the myofibroblast lineage, in favor of a more inflammatory phenotypic profile. To validate some of these results, we studied the expression of chemokines *CCL2* and *CCL7*, as well as the *PDGFRB* gene, in the human lung fibroblast cell line, IMR-90, after treatment with PDGFD (Supplementary Fig. 4a–c). These studies confirmed that the PDGFD signaling is functional in fibroblasts, as shown by downregulation of *PDGFRB* receptor expression, and that these potent chemokines can be upregulated in human fibroblast cells in the context of PDGFD stimulation.

Although endothelial cells (EC) exhibited a limited number of genes differentially regulated with *Pdgfd* KO, they did downregulate expression of leukocyte adhesion molecules *Icam1 and Selp*, inflammatory mediator *Il6st*, and the monocyte chemotactic factor gene *Ccl8*. (Supplementary Fig. 3c). Cells in the macrophage cluster showed down-regulation of a limited number of genes, including inflammatory mediators *Ccl8, Ccl12*, and *Ccr2*. This is consistent with the finding that they express low levels of Pdgf receptors (Supplementary Fig. 2e).

## Pdgfd promotes SMC phenotypic transition, expansion, and migration along with monocyte recruitment, but does not affect overall plaque burden

We next used histology methods to investigate the role of Pdgfd in atherosclerotic lesion features in the proximal aorta of the atherosclerosis mouse model. First, we employed X-gal staining to visualize the expression of the *lacZ* reporter gene integrated into the *Pdgfd* KO mouse genome[44]. We observed Xgal stained areas corresponding to SMC that comprise the medial layer of the aorta (Fig. 5a). Also, in the proximal aorta we observed patchy X-gal staining, and thus *Pdgfd* expression, in endothelial cells that lined the lumen of aortic regions with plaque. Interestingly, we also observed staining of endothelial cells lining the proximal coronary arteries, where SMC expression was not detected (Fig. 5a).

We also used histology to investigate the effect of Pdgfd deletion on atherosclerotic lesion anatomy in the aortic root by comparing KO and Ctl tissues after 16-week of HFD. The overall vessel area was not different between KO and Ctl vessels (Fig. 5b), and lesion area was not significantly different when compared to whole vessel (Fig. 5c) or medial area (Supplementary Fig. 5a). However, there was a significant decrease in acellular area (Fig. 5d). While the cellular mechanism underlying the origin of these regions is not clear, we have correlated lesion acellular area to SMC transition to the CMC phenotype, where these cells are localized in the plaque[16]. Importantly, we identified a highly significant decrease in total tdT lineage traced SMC in the vessel (Fig. 5e, f), and also in the plaque area (Supplementary Fig. 5b),

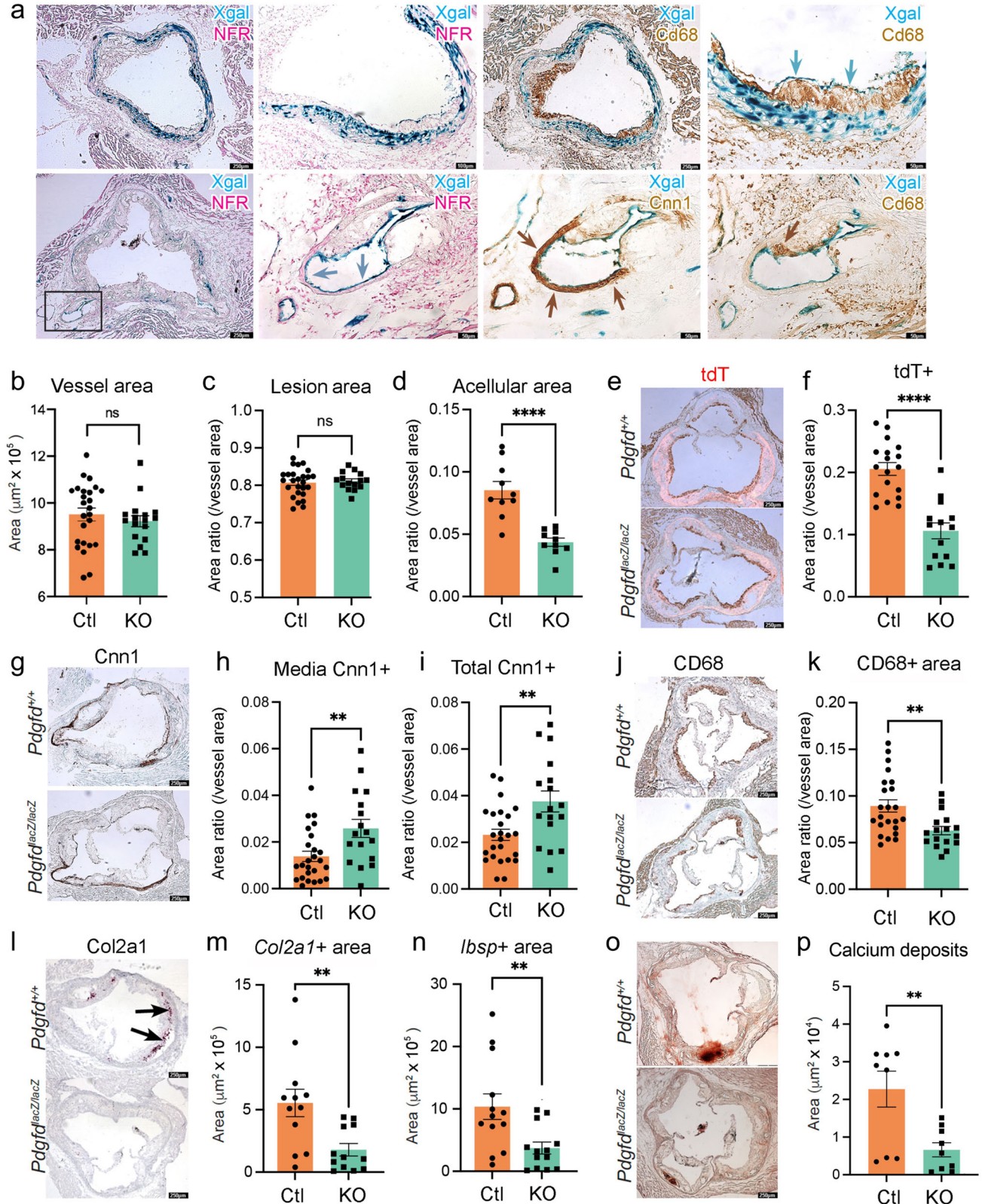

suggesting that Pdgfd is responsible for promoting expansion of the SMC lineage cells and their migration into the plaque. To quantify changes in SMC content in the different vessel compartments in the *Pdgfd* KO compared to Ctl mice, we performed Cnn1 staining and immunoreactive area measurements, and found significantly increased staining in the medial layer, as well as the overall area of KO vessels (Fig. 5g–i). These results are consistent with findings in the

scRNAseq data showing an increase in vessel number of differentiated contractile SMC. Plaque macrophage content as assessed with CD68 staining was decreased in the whole vessel (Fig. 5j, k) and specifically in the lesion area (Supplementary Fig. 5c) in the knockout mice, consistent with decreased monocyte recruitment. Also, expression of CMC markers *Col2a1* and *Ibsp* were significantly decreased in KO lesions compared to Ctl littermates (Fig. 5l–n), and this decrease

**Fig. 5 | In situ studies of mouse atherosclerosis reveal that Pdgfd KO lessens SMC cell state transitions and inflammation but without impact on plaque burden. a** X-gal staining visualizing β-galactosidase activity (lacZ, blue precipitate) to determine the cellular location of Pdgfd expression in mouse model atherosclerosis. Aortic root sections were also stained with a generic nuclear marker nuclear fast red (NFR), immunohistochemistry for the Cd68 macrophage marker or Cnn1 marker for SMC identification. **b** Quantification of total vessel area, *n* = 25 control and 17 KO mouse sections. **c** Quantification of lesion, *n* = 26 control and 17 KO mouse sections, and **d** acellular areas in Ctl and KO groups expressed as a ratio of the total vessel area per section, *n* = 10 control and 10 KO mouse sections, *p* < 0.0001. **e** Representative images identifying expression of the *tdTomato* gene to visualize the SMC lineage traced cells in aortic sections. **f** Quantification of *tdTomato* positive (tdT+) area relative to total vessel area, *n* = 18 control and 14 KO mouse sections, *p* < 0.0001. **g** Representative sections stained for Cnn1, a marker of the differentiated SMC. **h** Quantification of Cnn1 positive (Cnn1+) area at the media, *n* = 25 control and 17 KO mouse sections, *p* = 0.0061 and **i** compared to total cross-sectional area expressed as a ratio of the total vessel area per section, *n* = 25 control and 17 KO mouse sections, *p* = 0.0042. **j** Representative images of Cd68-stained aortic root area to quantify monocyte recruitment. **k** Quantification of Cd68 positive (Cd68+) area relative to the vessel area, *n* = 23 control and 17 KO mouse sections, *p* = 0.004. **l** Representative images of Col2a1 RNAscope of the aortic root in Ctl and KO mice. **m** Quantitative RNAscope of *Col2a1*, *n* = 12 control and 12 KO mouse sections, *p* = 0.0052 and **n** *Ibsp* expression, *n* = 13 control and 13 KO mouse sections, *p* = 0.0074. **o** Representative images stained for calcium deposits with alizarin red S. **p** Quantification of calcium deposits, *n* = 9 control and 9 KO mouse sections, *p* = 0.0076. Each dot represents quantification from identical level sections from individual animals. Data expressed as mean ± s.e.m with *p*-values using a two-sided unpaired t-test.

correlated to decreased aortic calcification as assessed with alizarin red S staining and quantification (Fig. 5o, p).

Taken together, these results support the scRNAseq findings and suggest two prominent mechanisms by which *Pdgfd* expression may promote disease risk. Pdgfd was found to promote de-differentiation of SMC, their migration into the plaque, and transition to the CMC phenotype, which we have correlated to disease risk[15,16]. Further, *Pdgfd* expression promotes monocyte-macrophage number in vascular lesions, presumably through recruitment, thus contributing to an inflammatory milieu. Surprisingly, these changes were not associated with a measurable effect on plaque burden.

## Blocking Pdgfd function in the mouse atherosclerosis model revealed disease related transcriptomic changes similar to those identified in the knockout model

To further investigate the transcriptomic effects of PDGFD in the disease setting, we treated the lineage tracing atherosclerosis *ApoE⁻/⁻* mouse model with a murine derived inhibitory monoclonal antibody directed against Pdgfd (25E17, PD-ab) or with a control IgG (Ctl-ab) (Fig. 6a). Its blocking activity was demonstrated with in vitro studies with human aortic smooth muscle cells, which showed decreased proliferation and migration in response to PDGFDD in the presence of antibody (Supplementary Fig. 6a, b). Our previous scRNAseq data indicated that *Pdgfd* RNA expression is low at baseline, and then increases with plaque progression and becomes prominent after 3 weeks of HFD feeding[12]. Therefore, we started administration of PD-ab in 11-week-old animals that had received 3 weeks of HFD and continued treatment until sacrificing animals after either 8 weeks exposure to the diet (5 weeks antibody) or 16 weeks diet (13 weeks antibody), and conducted scRNAseq at these timepoints, using identical methods to those described for the *Pdgfd* KO. Differential gene expression was identified using a cutoff value set to 0.05 for false discovery rate (FDR) *q*-value, and this analysis was conducted across all clusters because of the small number of differentially expressed genes in the SMC lineage clusters with Pdgfd blockade versus control antibody treatment (Supplementary Data 3).

A striking finding at the 8-week timepoint was the downregulation of genes in fibroblasts after 5 weeks of antibody treatment (Fig. 6b). These Fblst-1 genes which are upregulated by Pdgfd in the early disease setting included those related to extracellular matrix and migration (*Lama2, Smoc2*), proliferation and apoptosis (*Btg2, Akap12, Gadd45b*), endochondral bone formation and calcification (*Gdf10, Serpinf1, Clec3b*), chemotaxis (*Cxcl1, Cytl1*), and Pdgf signaling pathway (*Pdgfra*). Although not well represented in the heatmap, pericytes showed down-regulation of a number of immediate early genes including *Fos, Fosb, Junb, Ier2, Ier3*, and *Egr1*, suggesting an early effect on the response phenotype of these cells to Pdgfd stimulation. Upregulation of these genes would be expected if the differential expression was due to cellular stress conditions.

By 16 weeks of diet, there were extensive gene expression differences due to antibody blockade of Pdgfd protein function (Fig. 6c and Supplementary Data 3). The overall patterns of gene expression were similar to those identified with *Pdgfd* KO (Fig. 4d). Specific significant gene expression changes with Pdgfd Ab were most highly correlated with the knockout data for the CMC lineage phenotype, with 52 of the 89 Pdgfd Ab differentially downregulated genes showing significant decreased expression with Ab treatment. Pathways identified with antibody blockade of Pdgfd were also similar to those identified with the KO studies. FMC pathways were again identified as those supporting extracellular matrix and endochondral bone formation, and also showing enrichment for Tgfb-regulated genes (Supplementary Fig. 6c). CMC pathways were almost totally restricted to endochondral bone formation and ossification (Supplementary Fig. 6d). Pericyte genetic pathways identified with Pdgfd inhibition included those related to extracellular matrix organization and BMP signaling, with numerous different bone development pathways (Supplementary Fig. 6e). Inflammatory genes downregulated in the Fblst-1 cells were enriched in vascular disease peptidase pathways[49,50], and there was also enrichment for genes related to BMP signaling and bone development (Supplementary Fig. 6f).

Compared to Ctl-ab, PD-ab treatment induced suppression of SMC phenotypic transition as evidenced by a decreased relative number of CMC and increased number of SMC, and these effects were prominent after 16-weeks of HFD (Fig. 6d). The modest increase in FMC could be due to reduced transition to the CMC phenotype. Also, after 16 weeks of HFD, the PD-ab significantly reduced the number of pericytes in lineage traced cells and Fblst-1 cells in the non-lineage analysis (Fig. 6d, e). Surprisingly, the relative number of macrophages was found to be increased, but this was due in large part to the decrease in fibroblast and pericyte number in this type of analysis.

## Discussion

Signaling through the PDGF pathway is critical for the recruitment and expansion of mural vascular cell types during embryogenesis[26]. Renewed expression of PDGF ligands in the setting of disease has been linked to similar SMC cell state changes, but disease pathophysiology has not been ascribed to these functions. Although most recently discovered and least studied, *PDGFD* is the only PDGF pathway gene identified thus far in a CAD GWAS locus. The highly vascular cell-specific expression of this PDGF ligand and the fact that it binds the Pdgfrb receptor links it to the fundamental pathophysiology of atherosclerosis and specifically the contribution of the PDGF pathway to vascular wall cellular and molecular processes that promote CAD risk. In studies reported here, we have linked CAD GWAS association at 11q22.3 to *PDGFD* expression and have proposed a transcriptional mechanism for this association involving another putative CAD GWAS gene *FOXC1* that also has known regulatory roles in arterial development[51]. We have shown that expression of *Pdgfd* in experimental animal models mimics much of the same features that mark the

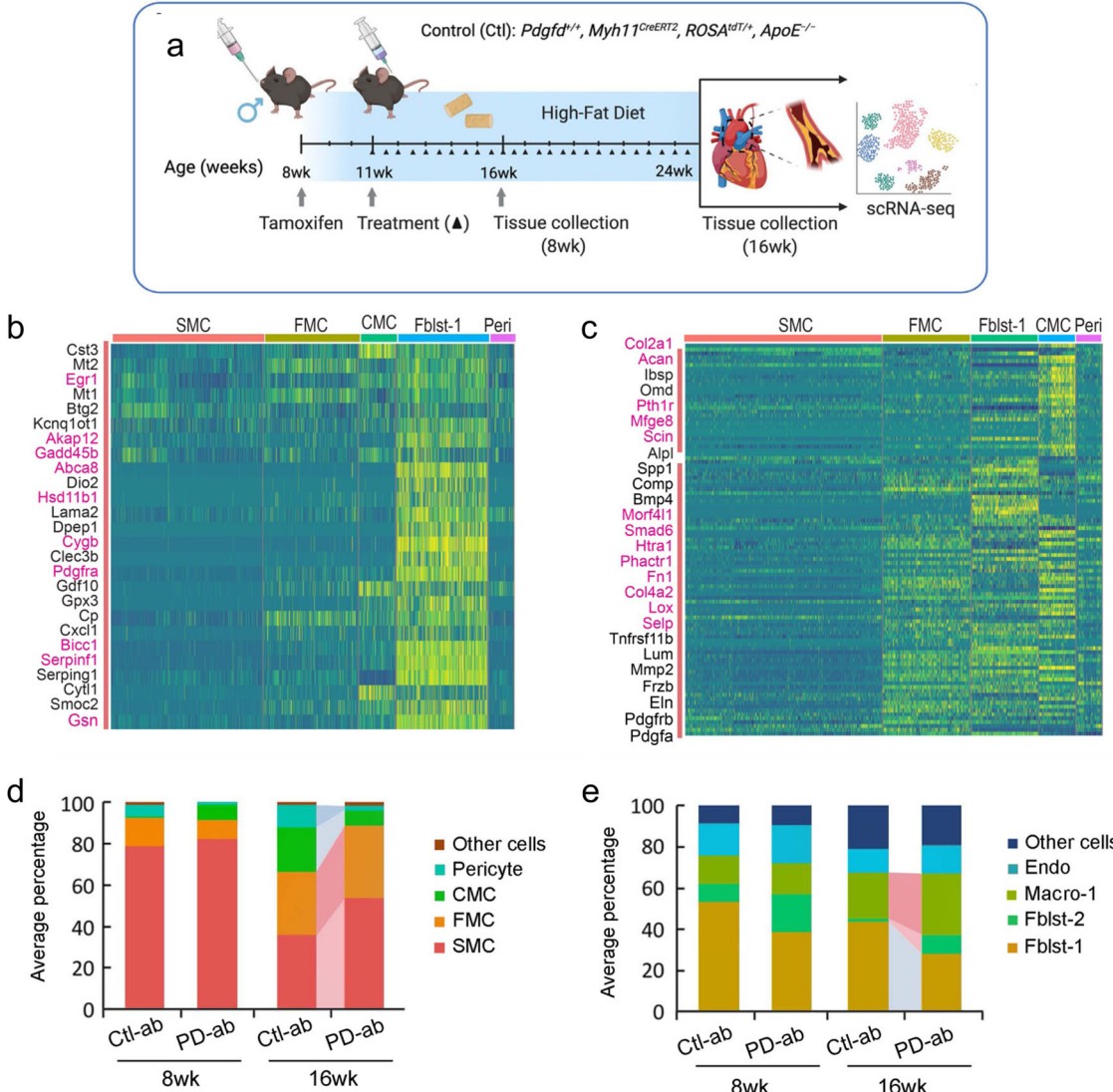

**Fig. 6 | Single cell RNA-seq studies of antibody-mediated Pdgfd inhibition in the mouse atherosclerosis model. a** Schematic of experimental design showing that SMC-specific lineage tracing wildtype mice were treated with tamoxifen at 8 weeks of age and tissues harvested after 8 and 16 weeks of high fat diet. Blocking Pdgfd antibody or isotype control antibody administration,10 mg/kg subcutaneously twice weekly, was initiated at 11 weeks and continued until animals were sacrificed after either 8 weeks exposure to the diet (5 weeks antibody) or 16 weeks diet (13 weeks antibody), and scRNAseq conducted at these timepoints. Created with BioRender.com. **b** Heatmap showing gene expression changes after 5 weeks of antibody treatment. The Fblst-1 cluster shows early downregulation of Pdgfd-regulated genes, and FMC and CMC cluster cells beginning to show evidence of upregulation of these genes as the SMC lineage cells are undergoing phenotypic transition in the developing lesion. Yellow color indicates differential down-regulated genes, genes in red text reside in window of lead SNP ± 500 kilobases. **c** Heatmap showing decreases in Pdgfd regulated genes across different cell clusters in targeted animals compared to controls. Yellow color indicates differential downregulated genes, genes in red text reside in window of lead SNP ± 500 kilobases. **d** Bar plot presenting the average percentage of lineage traced cells and **e** non-lineage traced cells in Ctl antibody and Pdgfd blocking antibody groups.

behavior of mural cell progenitors in embryonic development. In addition, in the disease setting Pdgfd produced by SMC and mural lineage cells promotes the expression of chemokines to promote the recruitment of inflammatory cells to the plaque.

Experiments reported here employing scRNAseq transcriptomic analysis showed that constitutive deletion of *Pdgfd* led to increased expression of SMC lineage genes and down-regulation of CMC gene expression profiles, as well as decreased numbers of cells that fit the FMC and CMC marker gene profiles[16]. Histology lesion analysis revealed that *Pdgfd* loss was associated with decreased total lineage traced cells in the vessel, decreased tdT positive cells in the lesion, and increased medial SMC number. Taken together, these data suggest that Pdgfd promotes SMC phenotypic modulation, proliferation, and migration into the plaque where SMC transition into modulated

phenotypes. Previous data from the Owens lab has shown that *Pdgfrb* SMC-specific deletion resulted in loss of the majority of SMC in the lesion[28]. While *Pdgfd* KO reveals a significant decrease in SMC migration into the plaque the phenotypic transition is obviously not as extensive, suggesting that Pdgfd contributes a substantial portion but not all of the migratory effect of Pdgf signaling, which is likely mediated through the Pdgfrb receptor. Despite these striking changes in vascular SMC phenotype, neither our studies of *Pdgfd* or the recent study of *Pdgfrb* showed a substantial effect on plaque burden in knockout mice for these two genes. Taken together these data suggest that SMC in general, and the PDGFD/PDGFRB signaling pathway, do not mediate CAD risk through altering the extent of disease but rather through regulation of disease features that regulate vascular stability.

The embryonic paradigm suggests that PDGF ligands produced by endothelial cells promote the contribution of smooth muscle progenitor cells to expansion, migration, and contribution to vascular development, and the effects of this pathway in the disease setting seem analogous. In mouse, *Pdgfd* is expressed at modest levels by Endo-1 cluster cells, at least in those cells associated with disease plaque, and *Pdgfrb* expressed by a majority of all SMC phenotype cells (Fig. 5a and Supplementary Fig. 2e). This is consistent with the scRNAseq and lesion histology data indicating that loss or blockade of Pdgfd function is associated with decreased numbers of SMC transition cells, and that *Pdgfd* deletion is associated with a decreased number of SMC lineage cells in the plaque and decreased overall SMC lineage number. Our data indicate that any endothelial chemotactic effect on SMC lineage cells is not entirely due to Pdgfd, as the loss of Pdgfrb results in a much greater decrease in SMC contribution to plaque than seen with knockout of *Pdgfd* in these studies. Interestingly, *Pdgfd* is also expressed by SMC and FMC, which may contribute to the expansion of the SMC lineage cells by an autocrine pathway, at least in the early stages of disease.

Despite the extensive evidence that inflammation is a key component of atherosclerosis, there has not been a large GWAS signal in loci harboring pro-inflammatory cytokines or chemokines, *IL6R* and *CXCL12* loci being notable exceptions. In studies reported here we documented a greater number of lesion macrophages in *Pdgfd* expressing compared to knockout animals, and this correlated with a more prominent expression of inflammatory genes in adventitial cells in the wildtype lesions. Expression differences for both adventitial fibroblasts and pericytes identified downregulated genes in the *Pdgfd* KO mice that were highly enriched for cytokine and chemokine chemotaxis mediators, with highly significant *p*-values and high relative numbers of represented genes per pathway. Compared to *Pdgfd* KO cells, wildtype pericytes were found to express genes related to leukocyte migration but also genes associated with FMC-like pathways such as extracellular matrix organization, and CMC-like pathways, including those related to ossification, suggesting that in the context of vascular disease processes and Pdgfd stimulation pericytes adopt a phenotypic modulation not unlike medial disease associated SMC. The inflammatory response to Pdgf signaling in mural cells has been reported previously, most notably in an elegant series of studies by Olson et al. with constitutive activation of the *Pdgfrb* receptor in transgenic mouse models[52]. Whether these gene expression changes in fibroblasts and pericytes reflect activation by Pdgfd emanating from the plaque cells or adventitial cells could not be addressed with this constitutive *Pdgfd* knockout model. The precise mechanisms by which chemokines expressed by adventitial cells might contribute to increased leukocyte trafficking to the plaque are not well understood, but a role for activated pericytes has been reported in the recruitment of immune cells to the vascular wall to promote inflammation and mitigate tumor growth[52,53]. Interestingly we saw only minimal changes in macrophage gene expression with *Pdgfd* KO, similar to findings in the *Smad3* KO where we did not see a significant number of DE genes with loss of *Smad3* expression[16]. Thus, the effect of this cell lineage on plaque anatomy may be primarily one of regulation of monocyte-macrophage number and not phenotype. We propose that *PDGFD* is an important contributor to the inflammatory cell milieu in the plaque, and that this mechanism accounts at least in part for its contribution to CAD risk.

SMC contribute the greatest portion of genetic attributable risk among all cells that participate in the vascular disease process, including endothelial, macrophage, T-cells, etc.[9]. For genome-wide significant genes that we have studied thus far, *TCF21*, *ZEB2*, *AHR*, and *SMAD3*, all inhibit disease risk. *PDGFD* is the first gene that we have studied that would promote disease risk, so comparison with other CAD disease genes expressed in SMC is important. The marked difference between the SMC transition program of these other CAD genes and *PDGFD* is that they promote transition primarily to the FMC phenotype or inhibit or fundamentally alter the CMC transition, while *PDGFD* promotes the SMC transition to cells that exhibit a CMC phenotype. An important corollary is that *Pdgfd* also promotes vascular calcification, e.g., while CAD risk inhibiting *Smad3* gene mitigates against vascular calcification. This is expected to contribute to detrimental features of plaque stability. Similar to *TCF21* and the other protective CAD genes, *PDGFD* also increases the transition to the FMC phenotype, but any beneficial effects of this function may be offset by an increase in the risk-related CMC phenotype. Further work is required to better understand the trajectories that usher cells into and through the FMC phenotype.

It is important to mention the limitations of these studies. While all human genetic data and a variety of fine mapping approaches point to rs2019090 and *PDGFD* as the disease CAD effectors at 11q22.3, there is considerable variation regarding which alleles at this locus promote disease risk and *PDGFD* expression. For instance, in early CAD GWAS studies the T allele was identified as promoting disease risk and *PDGFD* expression[29], while subsequent studies have primarily established that A is the CAD risk allele, and both GTEx and STARNET data reported as showing the A allele as also promoting *PDGFD* expression[31]. Despite GTEx eQTL findings reported here, our original analysis of STARNET data had identified the T allele as promoting *PDGFD* expression[30]. However, the accumulated human genetic data along with in vitro transcriptional studies reported here convincingly show that FOXC1/C2 binding at the A allele of rs2019090 promotes both disease risk and *PDGFD* expression. In addition, we point out the limitations of our constitutive *Pdgfd* knockout model, which may not fully or accurately reflect the in vivo role of this gene in disease, due to compensation during embryogenesis or in the disease environment. Importantly, while the Pdgfd blocking antibody studies provide evidence that the transcriptomic changes associated with knockout are representative of the proposed cellular effects of this gene in the disease setting, they do not provide direct evidence of disease causality. Nonetheless, the in vitro effects of the antibody toward SMC function suggest that this approach might be a rational in vivo method to prove causality and to investigate novel therapeutic avenues that blocking this pathway may provide. While beyond the scope of this study, further in-depth in vivo disease model studies investigating the effects of such blocking antibodies would be required to establish this possibility.

## Methods

### Colocalization analyses

Genomic location figures were generated in the UCSC Genome Browser. Visualization of CAD GWAS association and *PDGFD* expression quantitative trait loci was performed with locuscompare.com, as created by the Stephen Montgomery lab, Stanford. We conducted formal colocalization analysis using the fastEnloc method, with the meta-analysis results between CARDIoGRAMplusC4D and UK Biobank from van der Harst[6] and aortic artery eQTL from GTEx v8[35]. For GWAS data, we generated posterior inclusion probability (PIP) using torus. For eQTL, we used the precomputed PIP provided by fastEnloc. The GWAS and eQTL PIPs were used as input to fastEnloc for colocalization analysis. We selected a regional level colocalization probability (RCP) of 0.2 as the cutoff to select significant colocalization between the GWAS and eQTL.

### CRISPRi epigenome editing at rs2019090

Both dCas9-KRAB and single guide RNA sequences were cloned together into a lentiviral vector, virus packaged, and transduced into rs2019090 genotype AA homozygous HCASMC. After 6 h of virus infection, cells were refreshed with a complete medium and incubated for 3 days. RNA was then extracted, converted to cDNA, and analyzed using qPCR for expression of PDGFD and lncRNA *APOO2989.1*.

## Culture of human coronary artery smooth muscle cell (HCASMC)

Primary HCASMC were purchased from Cell Applications, Inc (San Diego, CA) and Lonza BioScience and were cultured in complete smooth muscle basal media (Lonza, #CC-3182) according to the manufacturer's instructions. All experiments were performed with HCASMC between passages 5–8. Rat aortic smooth muscle cells (A7r5) and human embryonic kidney 293 cells (HEK293) were purchased from ATCC and cultured in Dulbecco's Modified Eagle Medium (DMEM) high glucose (Fisher Scientific, #MT10013CV) with 10% FBS at 37 °C and 5% CO2. A7r5 at passage 6-18 were used for experiments. IMR90 fetal lung fibroblasts at passage 7 were cultured in FGM-2 lung fibroblast basal media (Lonza, #CC-3131) according to the manufacturer's instructions.

## Knockdown and over-expression

For the siRNA transfection, cells were grown to 30% confluence, then treated with siRNA or scramble control to a final concentration of 20 nM with RNAiMax (Invitrogen, Carlsbad, CA). The siRNAs for *PDGFD* were purchased from Origene (SR312885), and an equimolar combination of SR312885B and SR312885C employed. siRNA for *APO02989.1* was purchased from Dharmacon (SO-2964013G), and an equimolar combination of NGUTJ-000031 and NGUTJ-000033 were used for knockdown. Two different types of siRNAs for *FOXC1* and *FOXC2* were purchased from Thermo Fisher Scientific: Silencer (ASSAY ID #41733 for *FOXC1*, # s194416 for *FOXC2*), and Stealth (ASSAY ID #HSS142037 for *FOXC1* and #HSS142054 for *FOXC2*) reagents. Cells were treated with an equimolar combination of Silencer and Stealth and collected 72 h after transfection. For the overexpression study, viruses were produced with $8.5 \times 10^5$ HEK293T cells plated in each well of a six-well plate. The following day, plasmid encoding lentivirus was co-transfected with pMD2.G and pCMV-dR8.91 into the cells using Lipofectamine 3000 (Thermo Fisher, L3000015) according to the manufacturer's instructions. ViralBoost Reagent (AllStem Cell Advancements, VB100) was added (1:500) with fresh media after 5 h. Supernatant containing viral particles was collected 72 h after transfection and filtered. HCASMC were transduced with 2nd generation lentivirus with cDNAs cloned into pWPI (Addgene, 12254) using NEBuilder HiFi cloning (New England Biolabs). Cells were treated at 60% confluence with lentivirus for 5 to 24 h. The virus was removed and replaced with fresh media 48 h prior to collection for downstream applications. For PDGFD treatment study, IMR90 fibroblasts at 70–80% confluent were serum starved overnight and treated with 50 ng/ml of recombinant human PDGF-DD (R&D System, 1159-SB-025) for 24 h.

For evaluation of the effect of knockdown and over-expression of FOXC1/2 on the expression of endogenous genes, we normalized experimental results to the empty vector control results for each genotype group. This was necessary due to variation in the specific features of the individual genotype lines, such as transfection efficiency and transduction efficiency for these primary cultured human cells. All control values became one, and relative target expression thus determined. The t-test was performed between control and target within each genetic background for the reasons noted.

## RNA isolation and qRT-PCR

RNA was isolated using RNeasy plus micro kit (Qiagen, #74034) and total cDNA was prepared using High-capacity RNA-to-cDNA kit (Life Technologies, #4388950). Gene expression was assessed using TaqMan qPCR probes (Thermo Fisher) for *PDGFD* (Hs00228671_m1), *APO02989.1* (hs04980451_m1), *FOXC1* (Hs00559473_s1), *FOXC2* (Hs00270951_s1), *PDGFRA* (Hs00998018_m1), *PDGFRB* (Hs01019589_m1), *CCL2* (Hs00234140_m1), and *CCL7* (Hs00171147_m1) according to the manufacturer's instructions on a ViiA7 Real-Time PCR system (Applied

Biosystems, Foster City, CA). Relative expression was normalized to GAPDH (Thermo Fisher Sci., #4310884E) levels.

## Dual luciferase assays

FOXC1 or FOXC2 cDNAs were cloned into pWPI and transfect into A7R5 cells along with reporter constructs containing 3 copies of a 150 bp fragment encoding PDGFD locus sequence for the A allele (rs-2019090-A) or T allele (rs-2019090-T) at rs2019090. A7r5 cells were seeded into 24 well plate ($1.5 \times 10^4$ cells/well) in DMEM containing 10% FBS and incubated at 37 °C and 5% CO2 overnight. Cells were transfected with luciferase reporter plasmids (pLuc-MCS (empty), pLuc-Ax3, or pLuc-Tx3), cDNAs (pWPI (empty), pWPI-FOXC1 or pWPI-FOXC2), and *Renilla* luciferase plasmid using Lipofectamine 3000 (Invitrogen, #L3000015). Six hours after transfection, the media was changed to fresh complete media. Relative luciferase activity (firefly/*Renilla* luciferase ratio) was measured by SpectraMax L luminometer (Molecular Devices) 24 h after transfection. All experiments were conducted in triplicate and repeated at least 4 times.

## Mouse strains

For the unbiased fate mapping of SMCs with *Pdgfd* loss during disease progression, the *Pdgfd^lacZ/lacZ^* mouse strain obtained from Dr. Eriksson[44] was crossed with an SMC-specific lineage tracing *ApoE^−/−^* tandem dimer Tomato (tdT) fluorescent marker mouse model[12]. BAC transgenic mice that express a tamoxifen-inducible Cre recombinase driven by the SMC-specific *Myh11* promoter (*Tg^Myh11-CreERT2^*, JAX# 019079) were bred with a floxed tandem dimer tomato (tdT) fluorescent reporter line (B6.Cg-*Gt(ROSA)26Sor^tm14(CAGtdTomato)Hze^*/J, JAX# 007914) to allow SMC-specific lineage tracing. Mice were bred onto the C56BL/6, *ApoE^−/−^* background. Final genotypes of SMC lineage-tracing control (Ctl) mice were: *Pdgfd^+/+^*, *Myh11^CreERT2^*, *ROSA^tdT/+^*, *ApoE^−/−^*. Final genotypes of SMC lineage-tracing, *Pdgfd* KO mice were: *Pdgfd^lacZ/lacZ^*, *Myh11^CreERT2^*, *ROSA^tdT/+^*, *ApoE^−/−^*. As the Cre-expressing BAC was integrated into the Y chromosome, all lineage tracing mice in the study were male. The animal study protocol was approved by the Administrative Panel on Laboratory Animal Care (APLAC) at Stanford University.

## Induction of lineage marker by Cre recombinase

All mice received two doses of tamoxifen, at 0.2 mg/g^−1^ bodyweight, at a three day interval by oral gavage at 8 weeks of age to activate *Myh11-Cre*, before the HFD (Dyets, #101511, 21% anhydrous milk fat, 19% casein, and 0.15% cholesterol) was initiated.

## Mouse aortic root/ascending aorta cell dissociation

After 16 weeks of HFD for *Pdgfd* KO model experiments, or 8 weeks and 16 weeks HFD for PDGFD antibody experiments, animals were sacrificed and perfused with phosphate buffered saline (PBS). The aortic root and ascending aorta were excised, up to the level of the brachiocephalic artery, and washed three times in PBS. Collected tissues were placed into an enzymatic dissociation cocktail (2 U ml^−1^ liberase, Sigma–Aldrich #5401127001; 2 U ml^−1^ elastase, (Worthington, #LS002279) in Hank's Balanced Salt Solution (HBSS)) and minced. After incubation at 37 °C for 1 h, the cell suspension was strained, pelleted by centrifugation at $500 \times g$ for 5 min, and resuspended in fresh HBSS. For each scRNA capture, two mice were pooled as a group. Three and two separate pairs of isolation were performed for Ctl and *Pdgfd*-KO mice, respectively. Two separate pairs of isolations were performed for mice treated with control antibodies (Ctl-ab) or Pdgfd antibodies (PD-ab).

## FACS of mouse aortic root/ascending aorta cells

Cells were sorted by fluorescence-activated cell sorting (FACS), on a Sony SH800 instrument, based on tdTomato expression. tdT+ cells (considered to be of SMC lineage) and tdT− cells were captured on

separate but parallel runs of the same scRNAseq workflow, with gating strategy and threshold identical to those published in previous work[12] and datasets were later combined for all subsequent analyses. Representative FACS plots are shown in Supplementary Fig. 2a. Total cells were first separated from debris by plotting forward versus side scatter. Single cells (singlets) were then selected using forward scatter area (FSC-A) versus forward scatter height (FSC-H). From the singlet population, relevant cell populations were then selected by gating on tdT positive (tdT+) and tdT negative (Neg) events.

## Single cell capture and library preparation

All single cell capture and library preparation was performed at the Stanford Functional Genomics Facility (SFGF). Cells were loaded into a 10× Genomics microfluidics chip and encapsulated with barcoded oligo-dT-containing gel beads using the 10× Genomics Chromium controller according to the manufacturer's instructions. Single-cell libraries were then constructed according to the manufacturer's instructions. Libraries from individual samples were multiplexed into one lane prior to sequencing on an Illumina platform with targeted depth of 50,000 reads per cell.

## Preparation of mouse aortic root sections

Immediately after sacrifice, mice were perfused with 0.4% PFA. The mouse aortic root and proximal ascending aorta, along with the base of the heart, was excised and immersed in 4% PFA at 4 °C for 12 h. After passing through a sucrose gradient, tissue was frozen in OCT to make blocks. Blocks were cut into 7 μm-thick sections for further analysis.

## Immunohistochemistry and calcification assay

Slides were prepared and processed according to standard IHC protocol. Sections were incubated overnight at 4 °C with an anti-Cnn1 rabbit monoclonal primary antibody (1:400 dilution; TA327614; Origene), or a CD68 rabbit polyclonal antibody (1:300 dilution; ab125212; Abcam), after development with Dab, samples were mounted with EcoMount medium (Biocare Medical #EM897L). The processed sections were visualized using a Leica DM5500 microscope and images were obtained using Leica Application Suite X software. Areas of interest were quantified using ImageJ (NIH) software and compared using a two-sided t-test. Lesion size was defined by the area encompassing the intimal edge of the lesion to the border of Cnn1 positive intimal-medial junction. All area quantification was performed in a genotype blinded fashion with ImageJ using length information embedded in exported files. Cells near the caps were defined as cells within 30 μm of the lumen, as previously defined[13]. All biological replicates for each staining were performed simultaneously on position-matched aortic root sections to limit intra-experimental variance.

In situ assessment of lesion calcification in plaque sections was performed with 1% alizarin red S solution as per established protocol[54] and quantitation performed as described for immunohistochemistry studies.

## RNAscope assay

Slides were processed according to the manufacturer's instructions, and all reagents were obtained from ACD Bio (Newark, CA). Sections were incubated with commercially available probes against mouse *Col2a1* (#407221), *Ibsp* (#415501), or a negative control probe (#310043) for 2 h at 40 °C. Colorimetric assays were performed per the manufacturer's instructions.

## Analysis of scRNAseq data

Fastq files from each experimental group and mouse genotype were aligned to the reference genome (mm10) individually using CellRanger Software (10× Genomics). Individual datasets were aggregated using the CellRanger *aggr* command without subsampling normalization.

The aggregated dataset was then analyzed using the R package Seurat v4.1.1[55]. The dataset was trimmed of cells expressing fewer than 500 genes, and genes expressed in fewer than 50 cells. The number of genes, number of unique molecular identifiers and the percentage of mitochondrial genes were examined to identify outliers. As an unusually high number of genes can result from a 'doublet' event, in which two different cell types are captured together with the same barcoded bead, cells with >6000 genes were discarded. Cells containing >7.5% mitochondrial genes were presumed to be of poor quality and were also discarded. The gene expression values then underwent library-size normalization and normalized using established Single-Cell-Transform function in Seurat. Principal component analysis was used for dimensionality reduction, followed by clustering in principal component analysis space using a graph-based clustering approach via the Louvain algorithm. UMAPs were used for two-dimensional visualization of the resulting clusters. Analysis, visualization and quantification of gene expression and generation of gene module scores were performed using Seurat's built-in functions such as FeaturePlot, VlnPlot, DimPlot, DotPlot, DoHeatmap, FindMarkers, and AverageExpression. Heatmaps were generated with normalized data, based on top 40 differentially downregulated genes in individual clusters, except for the 8-week antibody treatment heatmap which was based on all differentially downregulated genes across all cells in that dataset. Putative CAD associated genes were identified as those residing in a window of lead SNP ± 500 kilobases, drawing association data from the recent Million Veterans Program data analyses[8]. DAVID/GSEA analyses were performed using a web-based platform at David.ncifcrf.gov and gsea-msigdb.org.

## Pdgfd blocking antibody generation and in vitro effects on human SMC

Mouse IgG1 anti-PDGFD monoclonal antibodies were generated by immunizing Pdgfd knockout mice with mature recombinant human PDGFD. Hybridoma clones were screened for binding antibodies to both human and mouse PDGFD. High affinity binders were screened for blocking PDGFD-mediated tyrosine 751 phosphorylation of the PDGFR beta expressed by mouse cardiac fibroblast, human osteosarcoma, and human aorta vascular smooth muscle cells. High potency blockers were screened for high selectivity over PDGFB and PDGFC binding. An isotype matched mouse IgG1 antibody that does not bind a mammalian protein served as a control antibody. Antibodies were formulated in 10 mM sodium acetate, 9% sucrose, pH 5.2, and administered at 10 mg/kg subcutaneously twice per week.

Pdgfd antibody blockade of SMC proliferation: Human aortic smooth muscle cells (HASMC) were plated at 5000 cells per well in 96-well plates and left to attach and spread for 30 h. Cells were then washed with serum free media 2 times and incubated in serum free media for 6 h. PDGF-DD (R&D systems, #1159-SB) was solubilized in 4 mM HCl (Vehicle). PDGF-DD (17.85 nM) was preincubated for 20 min with or without anti-PDGF-DD antibody 25E17 (35.7 nM) in serum free medium. Cells were treated with PDGF-DD with or without antibody in serum free media. Vehicle was added in control wells. Live cell proliferation was measured with the Incucyte Live-Cell Analysis system (Essen Bioscience). All data is expressed as the mean ± standard error of the mean (SEM). Statistical analysis performed using a repeated measures one-way ANOVA with Tukey correction.

Pdgfd antibody blockade of SMC migration: Human aortic smooth muscle cells (HASMC) were plated at 8000 cells per well in upper chamber of 96-well transmigration plates from Incucyte (cat # 4648) and left to attach and spread for 30 h. Cells were then washed with serum free media 2 times and incubated in serum free media for 12 hs. PDGF-DD (R&D systems, #1159-SB) was solubilized in 4 mM HCl (Vehicle). PDGF-DD (17.85 nM) was preincubated for 20 min with or without anti-PDGF-DD antibody 25E17 (35.7 nM) in serum-free

medium. PDGF-DD with or without antibody was added to the lower chamber of transmigration plates. Vehicle was added in control wells. Live cell migration was measured with the Incucyte Live-Cell Analysis system (Essen Bioscience). All data is expressed as the mean ± standard error of the mean (SEM). Statistical analysis performed using a repeated measures one-way ANOVA with Tukey correction.

### Statistics & reproducibility

All statistical analyses were conducted using GraphPad Prism software version 9. Difference between two groups were determined using an unpaired two-tailed *Student's t-test*. Differences between multiple groups were evaluated by one-way analysis of variance (ANOVA) followed by Dunnett's post-hoc test after the sample distribution was tested for normality. $P$ values <0.05 were considered statistically significant. All error bars represent standard error of the mean. Number of stars for the $P$-values in the graphs: *** $P < 0.001$; ** $P < 0.01$; * $P < 0.05$. No statistical method was used to predetermine sample size, which was based on extensive prior experience with this model. For single cell RNAseq of knockout animals, there was no randomization possible. All animals used for the antibody blocking were selected and grouped at random. The Investigators were blinded to sample genotype identity for the Pdgfd knockout lesion analysis, and for antibody blockade lesion analyses. The number of genes, number of unique molecular identifiers and the percentage of mitochondrial genes were examined to identify outliers in single cell analysis. As an unusually high number of genes can result from a 'doublet' event, in which two different cell types are captured together with the same barcoded bead, cells with >6000 genes were discarded. Cells containing >7.5% mitochondrial genes were presumed to be of poor quality and were also discarded. There were no other exclusions. All attempts of replication were successful.

### Reporting summary

Further information on research design is available in the Nature Portfolio Reporting Summary linked to this article.

## Data availability

Data generated through these studies have been uploaded to the Gene Expression Omnibus under accession code GSE214423. Source data are provided with this paper.

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

## Acknowledgements

The authors express their deepest gratitude to Dr. Eriksson at the Karolinska for providing the constitutive *Pdgfd* knockout mouse. In addition, we thank Haiyue Meng and Yujiao Luo for their help in making the colocalization plot, and Dr's Jae Lee, Aaron Winters, Chadwick King, and Chris Paszty for identification of the Pdgfd blocking antibody. This work was supported by National Institutes of Health grants K08HL153798 (P.C.), F32HL160067 (C.W.), L30HL159413 (C.W.), R01HL134817 (T.Q.), R01HL139478 (T.Q.), R01HL156846 (T.Q.), R01HL151535 (T.Q.), R01HL145708 (TQ), UM1 HG011972 (T.Q.), as well as a Human Cell Atlas grant from the Chan Zuckerberg Foundation. This work was also supported by a grant from the American Heart Association 20CDA35310303 (P.C.) and Amgen, Inc. (T.Q.).

## Author contributions

H.J.K. performed experiments and wrote the initial manuscript version; P.C. contributed to experimental design and manuscript; S.T. performed experiments; C.W. contributed to experimental design and manuscript; J.P.M. contributed to manuscript and data analysis; R.K. contributed to evaluation of knockout mouse phenotype; T.N. performed in vitro experiments; D.S. performed single cell analyses; H.S. helped with in vitro experiments; Y.L. conducted colocalization analysis; B.L. conducted colocalization analyses; S.H. contributed to experimental design and reagent development; S.J. contributed to experimental design and reagent development; T.Q. conceived of study, supervised experiments, helped with data analysis, and manuscript and figure preparation.

## Competing interests

C.S.W. is a consultant for Tensixteen Bio and Renovacor, T.Q. is a consultant for Saliogen, and a member of the Cardiometabolic Scientific Advisory Board of Amgen. S.J. and S.H. are paid employees of Amgen, and this work was supported in part by Amgen, Inc. All other authors declare no competing interests
