## [Peer Review File · Nature Communications]

Molecular mechanisms of coronary artery disease risk at the PDGFD locusReviewer #1 (Remarks to the Author):

The authors performed functional fine-mapping analyses at the disease susceptibility locus 11q22.3 for coronary artery disease and showed that the A allele of rs2019090 increased the PDGFD transcription level via promoting the FOXC1 transcription factor binding. To further investigate the PDGFD in the progression of coronary artery disease, they used atherosclerotic mouse models with conditional PDGFD KO to examine the effects in vascular tissues using single-cell RNA sequencing. The results showed that PDGFD promoted inflammatory cell mobilization and vascular calcification. To demonstrate the protective effects of PDGFD inhibition, they also performed experiments using PDGFD antibodies and obtained similar results. This is the first paper that the mechanism of PDGFD's involvement in coronary artery disease risk has been elucidated.

Comments

1. lines 58-60 "with further studies indicating that genes in these loci regulate the primary cellular processes that underlie the remaining disease risk through their effect on vascular wall cellular and molecular mechanisms"

Whereas the vessel wall is important, the disease-related organs identified in coronary artery disease GWAS are diverse, including the liver and adipose tissue. This point should also be mentioned.

2. lines 64-67 "it is increasingly clear that smooth muscle cells (SMC) confer a significant portion of the genetic disease risk"

Although the importance of SMCs is also mentioned, endothelial cells and fibers that makeup blood vessels are also considered to be important disease-related organs. The wording should be changed to take this into account.

3. Lines 104- Results, Fine mapping at the 11q22.3 CAD GWAS locus implicates rs2019090 as the causal variant and PDGFD as the causal gene

Regarding the fine mapping of the 11q22.3 locus, linking the locus with the PDGFD gene by eQTL or transcription factors predicted by motifs was good. On the other hand, a fine-mapping approach using linkage disequilibrium blocks was not conducted, which is usually performed as a GWAS downstream analysis. I suppose FINEMAP software or something like that can calculate posterior probabilities in the fine mapping analysis.

4. Lines 151-153. "We focused subsequent studies on FOXC1 since it resides in a CAD GWAS locus and combined eQTL and CAD association data suggest it is the causal gene in that locus (Fig.Suppl. Fig. 1C)"

Supple Fig. 1C alone did not reach a genome-wide significance level and was considered a weak basis for focusing on FOXC1. Please add more evidence to support it.

Incidentally, in the GWAS Catalog, FOXC1 is related to SBP, Waist-hip ratio, and Essential hypertension, and FOXC2 is related to Cortical thickness and White matter hyperintensity volume, so FOXC1 may indeed be more appropriate when considering atherosclerotic disease.

5. Fig2A-I

N=3 for each group was small. While you described that you chose three representative ones, what was the rationale for this? Additionally, please note what abbreviations for each sample means in the figure legend.

6. Fig2CD

You performed t-tests in those figures, but I don't know which samples to compare. Moreover, in the siRNA Knockdown experiment and the viral vector transfection experiment, the data should include an empty cassette as a control.

7. Lines166-168. "This difference likely represents the fact that knockdown studies query the endogenous chromatin environment while the reporter genes are not integrated into the genome and under epigenetic regulation"

You mentioned a chromatin environment here. Please provide the rationale for this. Or modify your argument appropriately.

8.Lines 169-170. "the structural AP002989.1 lncRNA, and eQTL studies have associated expression of this lncRNA with genotype at rs2019090"

I believe this is not eQTL but sQTL studies according to the GTEx database.

9.Lines 173-174 "However, we must note that the very low expression levels of this lncRNA make quantitation of changes in expression difficult to characterize."

There could be other ways of regulation than transcription factors. i.e. AP002989.1 might not be regulated via FOXC1.

10. Fig2J

What was the rationale for what was low titer, what was medium titer, and what was high titer in the virus experiments? Did high titers become cytotoxic? Did a secondary effect was observed in high-titer experiments?

11. Lines 206-209 "Both Pdgfd KO and control (Pdgfd+/+, Myh11CreERT2, ROSAtdT/+, ApoE-/-, designated as Ctrl) mice were administrated tamoxifen at the age of 8 weeks, followed by high-fat diet (HFD) feeding for 16 weeks to induce atherosclerosis"

When using such a model, it is necessary to quantitatively confirm the degree to which Knockout was actually achieved.

12. Line 213 "(two groups, two mice each group)"

Even though we are doing a single-cell analysis, did the data from just two mice provide sufficient evidence?

13. Lines231-236 "Interestingly Pdgfrb was high in all SMC lineage cells,..."

It was unclear whether there were statistically significant differences.

14 Lines237-243 "To examine changes in relative cluster cell numbers,..."

It was unclear whether there was statistical significance for these percentage alterations as well.

15 Lines 281-283 "Interestingly, when gene expression was compared across clusters at the individual cell level by heatmap, a significant overlap of gene expression between the FMC and Fblst-1 cluster was visualized (Fig. 4D)"

They might look similar, but the evaluation was not concrete. Please create a matrix and show the similarity in numbers, such as correlation coefficients.

16 Lines 284-285 "Down-regulated genes for the CMC transition phenotype were 4-fold greater in number than those identified for the FMC phenotype (Fig. 4C),"

The number of UP-regulated genes also increased about 4-fold, so it is necessary to include both Up- and Down-regulated genes in the discussion.

17 Lines 293-294 "included a greater number of putative CAD GWAS genes compared to other clusters, including Acan, Fn1, Mfge8, Phldb1, Smad7, Cd109, Eif4g2, and Nfat5"

Could you show us how enriched these genes are using an enrichment analysis?

18 Line 381 "0.05 for false discovery rate (FDR) q-value"

When you performed differential expression analyses, it is good to draw volcano plots, etc. with a log fold change threshold in addition to the q-value threshold to suppress false positive detections. What was the reason for not doing so?

Reviewer #2 (Remarks to the Author):

Several GWAS have identified a regulatory variant, rs2019090, in the PDGFD locus on chromosome 11q22.3 as a risk-associated allele in coronary artery disease (CAD). Studies showed that PDGFD transcription was promoted by FOXC1/C2 binding to the risk allele but not to the non-risk alleles. Further studies using single cell RNA sequencing using a *Pdgfd*^{-/-} mouse line revealed that *Pdgfd* promoted migration and expansion of smooth muscle cells (SMC) and a transition into a chondromyocyte phenotype associated with vascular calcification. An important finding was also that *Pdgfd*-expressing pericytes and fibroblasts had stronger expression of several chemokines and other inflammatory mediators suggesting an important role in the inflammatory response in developing plaques. However, genetic deletion of *Pdgfd*, as well as direct targeting of *Pdgfd* using a neutralizing antibody in an ApoE-deficient background surprisingly showed no association with fibrous cap formation or other measurable parameters of plaque burden. In conclusion, PDGFD appears to mediate the risk association of the identified allele through promotion of SMC expansion and migration connected to changes in SMC phenotypes, and to the promotion of an inflammatory response and increased leucocyte infiltration in the diseased vessel wall. This is a very nice study exploring the role at the CAD associated risk allele of PDGFD in CAD. The interpretations of the results are well founded and the manuscript is well written and easy to read despite the large sets of data presented.

Reviewer #3 (Remarks to the Author):

The manuscript by Kim et al. titled 'Molecular mechanisms of coronary artery disease risk at the PDGFD locus' studied the potential role of PDGF and its signaling on the development of atherosclerosis. The results show that the potential variant rs2019090 could affect the transcription factor FOXC1/C2 on the expression of PDGFD. The authors used constitutive *Pdgfd* knockout in the SMC lineage in an atherosclerotic mouse model and studied the single cell transcriptome and related the results to cell state and atherosclerotic lesions. They found that deletion of PDGFD could affect SMC expansion and migration and changes in lineage of SMC to chondromyocyte phenotype and possibly cellular calcification. In addition, PDGFD expressing fibroblasts and pericytes exhibit greater expression of chemokines and leukocyte adhesion molecules, which is consistent with the increase of macrophage recruitment in the plaques of these mice. However, these cellular changes did not affect the sizes of fibrous cap or the overall severity of the atherosclerotic burden. The authors further studied the effect of PDGF loss by using ApoE mice with an inhibitory monoclonal antibody directed against PDGFD, which had similar cellular findings as PDGF deletion mutant as described above. However, no pathologies of the treatment at 8 weeks or 16 weeks were provided using the inhibitory antibody of PDGFD.

Critique

This study by Kim et al. provided extensive and molecular mechanism information on the potential role of the locus rs2019090, which has been associated with coronary artery disease as a risk allele. Their data mainly derived in culture cells and the PDGFD deletion mice supported the idea that this risk allele is associated with PDGFD expression and transcription factor FOXC1/C2. The single cell analysis of the aorta of ApoE knockout mice with deletion of PDGFD also suggested that PDGFD can contribute to the migration, proliferation and possibly to the recruitment of inflammatory cytokines to arterial wall. These findings are strong to show that PDGFD plays a role in smooth muscle cell phenotype, which perhaps can contribute or enhance smooth muscle cell migration and enhancing inflammatory properties in the arterial wall of ApoE ^{-/-} mice.

However, there are multiple major concerns due to the author's frequent use of the term of causality attributed to the present findings. The results provided mainly suggest that this loci and PDGFD may contribute to the process of atherosclerosis, however there are

no direct data to demonstrate that their findings actually cause the severity of atherosclerosis since the PDGFD deletion mice had no changes in fibrous cap or severity of atherosclerosis compared to WT mice. No data on the extent necrotic lesions were provided. In addition, the finding of increased inflammatory cytokines also did not enhance the atherosclerotic process in the PDGFD KO mice.

1. The data using inhibitory monoclonal antibody to PDGFD is a very good approach and perhaps could have provided some strong data to support the potential causality involving PDGFD in the atherosclerotic process. However, it is surprising that the authors did not provide any analysis regarding the various aspects of measurement of atherosclerosis in ApoE -/- mice, such as lipid deposition, areas or counts of macrophages, extracellular matrix, smooth muscle cell number in the plaque, and sizes of necrotic lesions themselves. This clearly could be done and should have been done using 16-week treated mice. Thus, it would be very important for the authors to show that inhibitory Ab to PDGFD actually can change these findings in culture cells and changes in smooth muscle cell migration and proliferation can make a difference to the pathology of atherosclerosis.

2. If the authors cannot show a definitive effect of inhibitory PDGFD and changes in rs2019090 in the ApoE deletion mice, then the findings are very strong for biochemical and molecular analysis of the role of PDGFD and this loci, but they certainly do not strongly support their importance on causality of atherosclerosis, even in mouse model of atherosclerosis. Thus, for the present study, the authors need to remove most, if not all, the references to causality in their statements and provide more pathological data as described above.

REVIEWER COMMENTS - NCOMMS-22-35914 - “Molecular mechanisms of coronary artery disease risk at the PDGFD locus”

Reviewer #1 (Remarks to the Author):

The authors performed functional fine-mapping analyses at the disease susceptibility locus 11q22.3 for coronary artery disease and showed that the A allele of rs2019090 increased the PDGFD transcription level via promoting the FOXC1 transcription factor binding. To further investigate the PDGFD in the progression of coronary artery disease, they used atherosclerotic mouse models with conditional PDGFD KO to examine the effects in vascular tissues using single-cell RNA sequencing. The results showed that PDGFD promoted inflammatory cell mobilization and vascular calcification. To demonstrate the protective effects of PDGFD inhibition, they also performed experiments using PDGFD antibodies and obtained similar results. This is the first paper that the mechanism of PDGFD's involvement in coronary artery disease risk has been elucidated.

Comments

1. lines 58-60 “with further studies indicating that genes in these loci regulate the primary cellular processes that underlie the remaining disease risk through their effect on vascular wall cellular and molecular mechanisms”

Whereas the vessel wall is important, the disease-related organs identified in coronary artery disease GWAS are diverse, including the liver and adipose tissue. This point should also be mentioned.

Agreed. We have paraphrased the Reviewer comment in the revised manuscript, as indicated here.

Page 3, lines 56-60:

Genome wide association studies (GWAS) have identified hundreds of genomic loci that contribute to the genetic risk for CAD, with further studies indicating that genes in these loci regulate the primary cellular processes that underlie the remaining disease risk through their effect on vascular wall cellular and molecular mechanisms, as well as disease related processes in liver and adipose tissues^{1, 2, 3, 4}.

2. lines 64-67 “it is increasingly clear that smooth muscle cells (SMC) confer a significant portion of the genetic disease risk”

Although the importance of SMCs is also mentioned, endothelial cells and fibers that makeup blood vessels are also considered to be important disease-related organs. The wording should be changed to take this into account.

Agreed, we have edited the text to be more circumspect and to note the importance of other resident vascular cells in the disease process, as follows.

Page 3, lines 66-69:

While only a handful of these loci have been studied thus far, it is increasingly clear that smooth muscle cells (SMC), endothelial cells (EC) and macrophages confer a significant portion of the genetic disease risk, through phenotypic transitions that are mediated by dramatic cell state changes.

3. Lines 104- Results, Fine mapping at the 11q22.3 CAD GWAS locus implicates rs2019090 as the causal variant and PDGFD as the causal gene

Regarding the fine mapping of the 11q22.3 locus, linking the locus with the PDGFD gene by eQTL or transcription factors predicted by motifs was good. On the other hand, a fine-mapping approach using linkage disequilibrium blocks was not conducted, which is usually performed as a GWAS downstream analysis. I suppose FINEMAP software or something like that can calculate posterior probabilities in the fine mapping analysis.

Fine-mapping in this locus has been published in association with coronary artery disease (CAD) GWAS meta-analyses by other groups who have shown that one or another of the SNPs in the CAD haploblock is linked to PDGFD¹. Also, SNPs in the disease haploblock at 11q22.3 have been linked to PDGFD by Hi-C and eQTL

data in relevant tissues ⁴. While various SNPs in this locus have been identified as the disease variant, studies are confounded by high linkage disequilibrium (LD) with all variants being in complete LD with rs2019090, the SNP that was implicated in our previous genomic studies and further validated as the causal SNP in the mechanistic studies reported in this manuscript.

However, to further validate that *PDGFD* is the primary mediator of disease risk in this locus, we have investigated this question with CRISPRi methodology. We directed three single guide RNAs to rs2019090, and transfected them into primary cultured smooth muscle cells (HCASMC) along with an epigenetic modulating dCas9KRAB expression construct. This approach directs epigenetic silencing to a restricted window ~1 kb around rs2019090. These studies are now presented in Figures 1F and 1G of the manuscript and show that all of the sgRNAs reduced expression of *PDGFD*, and interestingly did not suppress expression of the lncRNA. In conjunction with the fine mapping, this epigenetic editing of the genome provides definitive proof that *PDGFD* is the gene that drives association at the locus under study. The CRISPRi studies are presented in the Results section of the text, as shown here, and methods described in other relevant sections.

Page 6, line 145-155:

To experimentally investigate whether *PDGFD* is the disease related gene at 11q22.3, we employed epigenetic targeting at the rs2019090 variant. CRISPRi was conducted by transducing an HCASMC line with the AA genotype, line 2897, with lentiviruses encoding dCas9KRAB along with one of three single guide RNAs (Suppl. Fig. 1C). Gene expression was evaluated by quantitative real-time PCR, for both *PDGFD* and the lncRNA *AP002989.1* (Fig. 1F, G). These experiments indicated that *PDGFD* expression was highly significantly suppressed by all three guides, but interestingly the lncRNA expression was not affected. It is a consideration that CRISPRi with this approach suppresses over a distance of 1-2 kilobases, but there are no other protein coding genes within 100,000 base pairs of the targeted region. These findings support the identification of *PDGFD* as the disease associated gene and indicate that lncRNA *AP002989.1* is not a direct target of the disease association mechanism.

4. Lines 151-153. “We focused subsequent studies on *FOXC1* since it resides in a CAD GWAS locus and combined eQTL and CAD association data suggest it is the causal gene in that locus (Fig. Suppl. Fig. 1C)”
Supple Fig. 1C alone did not reach a genome-wide significance level and was considered a weak basis for focusing on *FOXC1*. Please add more evidence to support it. Incidentally, in the GWAS Catalog, *FOXC1* is related to SBP, Waist-hip ratio, and Essential hypertension, and *FOXC2* is related to Cortical thickness and White matter hyperintensity volume, so *FOXC1* may indeed be more appropriate when considering atherosclerotic disease.

We thank the Reviewer for this helpful comment. Both *FOXC1* and *FOXC2* have reached genome-wide significance in the recent MVP CAD GWAS meta-analysis, and both are included in the 95% credible sets for CAD association. However, as noted by the Reviewer, *FOXC1* is also associated with vascular risk factors including hypertension, systolic blood pressure, and waist hip ratio, while *FOXC2* is associated with non-vascular traits and diseases, including heel mineral density, cortical thickness and white matter hyperintensity volume. Also importantly, both genes have shown the same effect on transcription in the luciferase reporter gene studies and other in vitro studies. We have expanded the rationale for focusing on *FOXC1* as suggested by the Reviewer, and eliminated the graph showing co-localization of the QTL and CAD association (previous Suppl. Fig. 1C), as these data are clearly out of date and do not represent a formal analysis of *FOXC1* CAD association.

Page 7, lines 169-176:

While both of these TFs reside in CAD associated loci, and thus may be directly linked to CAD risk, we have decided to focus subsequent studies on *FOXC1*. The transcriptionally active A allele is more highly represented in its binding sites, *FOXC1* mutations have been linked to PDGF signaling in the context of cerebral small vessel disease, and this gene has also been linked to vascular risk factors including hypertension, systolic blood pressure, and waist hip ratio (GWAS catalog). *FOXC2* has been related primarily to non-vascular phenotypes, including cortical thickness and white matter hyperintensity volume (GWAS catalog).

5. Fig2A-I, N=3 for each group was small. While you described that you chose three representative ones, what was the rationale for this? Additionally, please note what abbreviations for each sample means in the figure legend.

Yes, we agree these analyses are not clear from the text. We conducted 4 separate experiments, with each experiment having three biological replicates. All experiments gave similar results, and one experiment was randomly chosen to be shown in the figures. The three datapoints are shown along with the results from the unpaired *t*-test. To show the consistency of data from these in vitro assays, we are showing here all the data collected for the luciferase assays shown in Fig. 2A and 2B. We have analyzed the mean from the 3 replicates for each of the four individual experiments and graphed the results (**Reviewer Figs. 1, 2**). We have added text to clarify the methods, and also we have now clarified the abbreviations in the figure legend.

Page 7, lines 165-166:

These and other in vitro assays were performed at least three times with each having at least three biological replicates.

6. Fig2CD

You performed *t*-tests in those figures, but I don't know which samples to compare. Moreover, in the siRNA Knockdown experiment and the viral vector transfection experiment, the data should include an empty cassette as a control.

We apologize for the confusion. For these analyses, to accommodate all of the data in one graph, we normalized the experimental results to the respective control results (empty lentivirus vector control or siRNA control) for each genotype group. This was necessary due to variation in the specific features of the individual genotype lines, such as transfection efficiency and transduction efficiency for these primary cultured human cells. All control values became one, and relative target expression thus determined. The *t*-test was performed between control and target within each genetic background for the reasons noted. These results showing only the normalized target data and *p*-values are presented in the final graph for presentation purposes. We now describe the important aspects of the data analysis in more detail in the Methods section of the revised manuscript.

Page 23, lines 600-607:

For evaluation of the effect of knockdown and over-expression of FOXC1/2 on expression of endogenous genes, we normalized experimental results to the respective control results (empty lentivirus vector control or siRNA control) for each genotype group. This was necessary due to variation in the specific features of the individual genotype lines, such as transfection efficiency and transduction efficiency for these primary cultured human cells. All control values became one, and relative target expression thus determined. The *t*-

test was performed between control and target within each genetic background for the reasons noted. These results, showing only the normalized target data and p-values, were presented in the final graph for presentation purposes.

7. Lines 166-168. *“This difference likely represents the fact that knockdown studies query the endogenous chromatin environment while the reporter genes are not integrated into the genome and under epigenetic regulation”*You mentioned a chromatin environment here. Please provide the rationale for this. Or modify your argument appropriately.

This comment was intended to provide a possible explanation why the luciferase reporter experiments showed FOXC1 suppression of T-reporter gene transcription at rs2019090 (**Fig. 2A**) while it did not suppress expression of the endogenous PDGFD gene (**Fig. 2C**). Our intention was to point out that one difference between the two types of experiments is that the reporter genes are not under epigenomic control. However, there is no evidence that this is the case, and is only one of many possible explanations, and for that reason we have now removed this sentence.

8. Lines 169-170. *“the structural AP002989.1 lncRNA, and eQTL studies have associated expression of this lncRNA with genotype at rs2019090.”* I believe this is not eQTL but sQTL studies according to the GTEx database.

We thank the Reviewer for catching this mistake, we have now corrected the text in the revised manuscript.

Page 8, lines 188-190:

..the structural AP002989.1 lncRNA, and eQTL studies have associated **splicing** of this lncRNA with genotype at rs2019090, we performed similar studies examining the effects of FOXC1 perturbation on expression of this gene.

9. Lines 173-174 *“However, we must note that the very low expression levels of this lncRNA make quantitation of changes in expression difficult to characterize.”*

There could be other ways of regulation than transcription factors. i.e. AP002989.1 might not be regulated via FOXC1.

We are not exactly certain regarding the point made by the Reviewer here, but we assume that it is in regard to the speculation whether the lncRNA is or is not involved in the genetic CAD association at this locus. While we do show that there is some functional interaction between the lncRNA and PDGFD gene (Fig. 2G, L), we now provide (**Fig. 1F, 1G**) definitive CRISPRi data that PDGFD and not the lncRNA is the target of allelic variation at this locus. We have thus now elected to strike this sentence from the revised manuscript.

10. Fig 2J -*What was the rationale for what was low titer, what was medium titer, and what was high titer in the virus experiments? Did high titers become cytotoxic? Did a secondary effect was observed in high-titer experiments?*

We apologize for the confusing text. The overall goal of these studies was to investigate the interactions among PDGFD, AP002989.1, FOXC1 and PDGF receptors, by altering PDGFD expression, with the dose response being used to verify specificity of effect. The experiments were performed as follows. Several different batches of viral transduced cells were evaluated by qPCR for PDGFD expression, and batches divided into tertiles on the basis of expression level. The differences in batch to batch variation was likely a stochastic function of the number of viral integrations in the cells, due to viral titer among other variables. We used this variability among batches to identify the dose response to PDGFD expression levels. We have extensive experience with lentivirus transduction and we saw no evidence of toxicity which is commonly identified by floating cells and other measures of decreased viability. We have edited the text to be more clear regarding the method of generating variable PDGFD expression in the three groups of cells.

Page 8, lines 202-205:

We grouped batches of HCASMC expressing varying levels of PDGFD after transduction with lentivirus, dividing them into tertiles for low, moderate and high expression levels, and used quantitative RT-PCR to study the transcriptional response of related factors to increased PDGFD.

11. Lines 206-209 “Both *Pdgfd* KO and control (*Pdgfd*^{+/+}, *Myh11CreERT2*, *ROSAtdT*^{+/+}, *ApoE*^{-/-}, designated as *Ctl*) mice were administrated tamoxifen at the age of 8 weeks, followed by high-fat diet (HFD) feeding for 16 weeks to induce atherosclerosis.”
When using such a model, it is necessary to quantitatively confirm the degree to which Knockout was actually achieved.

Yes, one should always confirm that the experimental mouse line is actually missing the functional gene and missing transcripts from the targeted gene. The gene deletion and loss of the *Pdgfd* transcript were confirmed in the original Swedish study by scientists who generated the mouse (Reviewer Fig. 3)⁵. Gene targeting on *Pdgfd* gene expression can be seen in Suppl. Table 2, which shows that *Pdgfd* is the most highly downregulated gene in knockout cells where it is normally expressed in the vascular wall, including smooth muscle cells, transition fibromyocytes, pericytes and fibroblast cell populations. In addition, we confirmed by PCR genotyping assays that every mouse included in these studies carried two *Pdgfd* targeted alleles. Finally, individual reads for each exon are shown for individual captures in the following table, and indicate that the number of reads is at a background level for all 7 exons of the *Pdgfd* gene in the KO condition.

Evaluation of *Pdgfd* knockout, scRNAseq Reads are shown per exon, for both SMC lineage and non-SMC lineage cells, 16 wks high fat diet.

Exon #	Control_tdT	Control_non-tdT	Pdgfd _KO_tdT	Pdgfd _KO_non-tdT
1	1451	652	21	26
2	1563	480	6	6
3	1038	374	7	7
4	381	130	1	6
5	521	181	1	7
6	547	152	3	9
7	4562	1293	16	17

12. Line 213 “(two groups, two mice each group)”

Even though we are doing a single-cell analysis, did the data from just two mice provide sufficient evidence?

This study was actually composed of 26 mice, including 14 experimental and 12 control animals: 6 KO and 4 control animals at 16 wks high fat diet, and for the Ab studies 4 *Pdgfd* Ab treated mice and 4 control Ab treated mice times two because of two timepoints (see Table). Data from all mice was processed and aggregated together, and we would have easily identified aberrations in cell capture or processing. The results from the KO and Ab treated groups were highly similar, as described in the manuscript, so altogether, as scRNAseq studies go, this was a robust study with a built-in validation design that has not been accomplished previously.

Tdt(+)/Tdtneg (-)	8 wk	16 wk
Ctl (control)-Ab	2 captures x 2 mice/capture	2 captures x 2 mice/capture
PDGFD-Ab	2 captures x 2 mice/capture	2 captures x 2 mice/capture
Pdgfd - KO		3 captures x 2 mice/capture
Pdgfd control		2 captures x 2 mice/capture

13. Lines 231-236 “Interestingly *Pdgfrb* was high in all SMC lineage cells,…” It was unclear whether there were statistically significant differences.

Sorry for the confusion. By making this statement we were simply noting that all SMC lineage cells expressed the receptor and thus could respond to *Pdgfd* secreted by the SMC themselves (autocrine stimulation) or another adjacent cell (juxtacrine signaling). There was no intention to make specific comparisons between *Pdgfrb* expression by different cellular lineages. We have edited the text to state “Interestingly, *Pdgfrb* was expressed in all SMC lineage cells.”

14 Lines 237-243 “To examine changes in relative cluster cell numbers,…” It was unclear whether there was statistical significance for these percentage alterations as well.

We are not aware of a statistical test that can examine the statistical significance of the differences in mean values of the large cell cluster numbers. This appears to be a classical “large p, small n” situation and thus a high-dimensional data design. To validate the changes in cluster cell numbers identified with single cell data, we have performed histological evaluation of cell numbers for the macrophages (CD68+ area), CMC (Col2a1+, Ibsp+ area), and SMC numbers (Cnn1+ area). In addition, we have validated the most important findings from the *Pdgfd* KO scRNAseq data with a second scRNAseq dataset, obtained with the *Pdgfd* Ab knockdown study.

15 Lines 281-283 “Interestingly, when gene expression was compared across clusters at the individual cell level by heatmap, a significant overlap of gene expression between the FMC and *Fblst-1* cluster was visualized (Fig. 4D)”. They might look similar, but the evaluation was not concrete. Please create a matrix and show the similarity in numbers, such as correlation coefficients.

The Reviewer is correct, there is no enrichment at the individual gene level, despite the apparent overlap of DE genes in the heatmap. This text has been removed from the Results section of the revised manuscript.

16 Lines 284-285 “Down-regulated genes for the CMC transition phenotype were 4-fold greater in number than those identified for the FMC phenotype (Fig. 4C),” The number of UP-regulated genes also increased about 4-fold, so it is necessary to include both Up- and Down-regulated genes in the discussion.

We thank the Reviewer for this comment. We had performed pathway analysis with the upregulated gene list, however there were no clearly identified pathways that informed on *Pdgfd* function or disease mechanisms. These included “multicellular organism process” and “developmental process” among other general terms. We now mention this finding in the Results section of the manuscript.

Page 12, line 310-314:

There were an equally large number of downregulated genes in the *Pdgfd* knockout compared to wildtype CMC clusters, but these genes were not enriched in pathways that were informative regarding *Pdgfd* function or disease mechanisms. Pathways identified included “multicellular organism process” and “developmental process” among other general terms.

17. Lines 293-294 “included a greater number of putative CAD GWAS genes compared to other clusters,

including *Acan*, *Fn1*, *Mfge8*, *Phldb1*, *Smad7*, *Cd109*, *Eif4g2*, and *Nfat5*”

Could you show us how enriched these genes are using an enrichment analysis?

We ran a Fisher’s exact test and there was no difference in enrichment between the different SMC clusters. We were misled because the genes which appeared in the CMC cluster were more easily recognized as likely disease risk modulating genes. At any rate, thank the Reviewer for this comment and we have removed this statement from the manuscript.

18 Line 381 “0.05 for false discovery rate (FDR) q-value” When you performed differential expression analyses, it is good to draw volcano plots, etc. with a log fold change threshold in addition to the q-value threshold to suppress false positive detections. What was the reason for not doing so?

The default setting for the non-parametric Wilcoxon rank sum test in Seurat is ~1.4 fold change cutoff, and for the p-value Bonferroni correction is used to calculate adjusted p-value cut off =0.05. This is the case for all analyses in this study and indeed is the default for data analysis in this lab. We have made many volcano plots in the early days of Seurat directed differential gene analysis but have never found them to point out a problem with the analysis.

Reviewer #2 (Remarks to the Author):

Several GWAS have identified a regulatory variant, rs2019090, in the PDGFD locus on chromosome 11q22.3 as a risk-associated allele in coronary artery disease (CAD). Studies showed that PDGFD transcription was promoted by FOXC1/C2 binding to the risk allele but not to the non-risk alleles. Further studies using single cell RNA sequencing using a *Pdgfd*^{-/-} mouse line revealed that *Pdgfd* promoted migration and expansion of smooth muscle cells (SMC) and a transition into a chondromyocyte phenotype associated with vascular calcification. An important finding was also that *Pdgfd*-expressing pericytes and fibroblasts had stronger expression of several chemokines and other inflammatory mediators suggesting an important role in the inflammatory response in developing plaques. However, genetic deletion of *Pdgfd*, as well as direct targeting of *Pdgfd* using a neutralizing antibody in an ApoE-deficient background surprisingly showed no association with fibrous cap formation or other measurable parameters of plaque burden. In conclusion, PDGFD appears to mediate the risk association of the identified allele through promotion of SMC expansion and migration connected to changes in SMC phenotypes, and to the promotion of an inflammatory response and increased leucocyte infiltration in the diseased vessel wall. This is a very nice study exploring the role at the CAD associated risk allele of PDGFD in CAD. The interpretations of the results are well founded and the manuscript is well written and easy to read despite the large sets of data presented.

Reviewer #3 (Remarks to the Author):

The manuscript by Kim et al. titled ‘Molecular mechanisms of coronary artery disease risk at the PDGFD locus’ studied the potential role of PDGF and its signaling on the development of atherosclerosis. The results show that the potential variant rs2019090 could affect the transcription factor FOXC1/C2 on the expression of PDGFD. The authors used constitutive Pdgfd knockout in the SMC lineage in an atherosclerotic mouse model and studied the single cell transcriptome and related the results to cell state and atherosclerotic lesions. They found that deletion of PDGFD could affect SMC expansion and migration and changes in lineage of SMC to chondromyocyte phenotype and possibly cellular calcification. In addition, PDGFD expressing fibroblasts and pericytes exhibit greater expression of chemokines and leukocyte adhesion molecules, which is consistent with the increase of macrophage recruitment in the plaques of these mice. However, these cellular changes did not affect the sizes of fibrous cap or the overall severity of the atherosclerotic burden. The authors further studied the effect of PDGF loss by using ApoE mice with an inhibitory monoclonal antibody directed against PDGFD, which had similar cellular findings as PDGF deletion mutant as described above. However, no pathologies of the treatment at 8 weeks or 16 weeks were provided using the inhibitory antibody of PDGFD.

Critique

This study by Kim et al. provided extensive and molecular mechanism information on the potential role of the locus rs2019090, which has been associated with coronary artery disease as a risk allele. Their data mainly derived in culture cells and the PDGFD deletion mice supported the idea that this risk allele is associated with PDGFD expression and transcription factor FOXC1/C2. The single cell analysis of the aorta of ApoE knockout mice with deletion of PDGFD also suggested that PDGFD can contribute to the migration, proliferation and possibly to the recruitment of inflammatory cytokines to arterial wall. These findings are strong to show that PDGFD plays a role in smooth muscle cell phenotype, which perhaps can contribute or enhance smooth muscle cell migration and enhancing inflammatory properties. The results provided mainly suggest that this loci and PDGFD may contribute to the process of atherosclerosis, however there are no direct data to demonstrate that their findings actually cause the severity of atherosclerosis since the PDGFD deletion mice had no changes in fibrous cap or severity of atherosclerosis compared to WT mice. in the arterial wall of ApoE -/- mice.

However, there are multiple major concerns due to the author's frequent use of the term of causality attributed to the present findings. No data on the extent necrotic lesions were provided. In addition, the finding of increased inflammatory cytokines also did not enhance the atherosclerotic process in the PDGFD KO mice.

We thank the Reviewer for their positive comments, and apologize for the use of the term “causality.” This word had been deleted entirely from the manuscript, and the findings reported regarding the mouse phenotype have been interpreted in a more balanced and reserved fashion. However, we would respectfully submit that the lesion phenotypes that were characterized in the mouse KO, and to a lesser extent the antibody treated lesions, could reasonably be expected to provide the mechanism(s) by which variation at the *PDGFD* locus is responsible for disease risk. We would note the human disease feature primarily studied by GWAS in the human cohorts that constitute CARDIoGRAM and the million veterans program is myocardial infarction or symptoms and signs of CAD that are directly linked to MI. Further, it has been described that in a significant number of cases that rupture occurs in lesions that represent moderate occlusion of 40-60 % of the vessel diameter^{6, 7, 8}. General inflammatory profile, macrophage content, calcification, and smooth muscle cell phenotypic transitions have all been posited as high probability mechanisms by which plaque is destabilized to promote MI,^{9, 10, 11, 12, 13} and we have documented *Pdgfd* effects on all of these disease features. Thus, while plaque burden in the hyperlipidemic mouse has historically been felt to provide a good readout on risk through measures of plaque burden, it would seem possible that not all CAD gene disease phenotypes are working through increased plaque burden. Further, deletion of two of the most validated CAD loci that we have studied, *TCF21* and *ZEB2*, did not show a change in plaque burden in the ApoE KO model. As noted by the Reviewer, fibrous cap thickness is also a relevant variable in considering risk for plaque rupture. We did not find a significant difference in the SMC area at the fibrous cap, but there was small increase in mean cap SMC area in the *Pdgfd* KO animals, 0.0023 vs 0.0017 μm^2 , $p=0.18$) (**Reviewer Fig. 4**). Also, the phenotype of SMC derived cells at the cap may be important in the relative risk of plaque rupture, as well as the area. We know very little about the origin of the cap SMC and how their phenotype might modulate CAD risk, and we do not even know whether transition SMC lineage cells at the cap express contractile markers and would thus be included in measures of cap thickness.

Regarding quantification of “necrotic lesion” area, we are not certain exactly which lesion measurement the Reviewer is referencing. There is unfortunately some confusion regarding the terminology but it seems most authors use the terms necrotic core and lipid core interchangeably to refer to the primary lipid rich and cell depleted body of the lesion. In that regard, we did quantify the lipid rich lesion area, which was not changed between the different genotypes, as noted by the Reviewer. “Necrotic area” or “acellular area” also appears to be used to describe acellular areas at the base of the plaque, in juxtaposition to the media. In this manuscript we refer to these as “acellular areas” and have indeed quantified these areas as well and shown that they are decreased in the *Pdgfd* KO lesions (**manuscript Fig. 5D**). We have noted in the context of other gene knockouts that the basilar acellular plaque area is highly correlated with SMC transition to the CMC phenotype, with increasing area in the animal models with greater chondromyocyte burden. We have identified the CMC to

be in close juxtaposition to these acellular areas and for these acellular areas to directly correlate to CMC number and vascular calcification in our knockout studies.

1. *The data using inhibitory monoclonal antibody to PDGFD is a very good approach and perhaps could have provided some strong data to support the potential causality involving PDGFD in the atherosclerotic process. However, it is surprising that the authors did not provide any analysis regarding the various aspects of measurement of atherosclerosis in ApoE -/- mice, such as lipid deposition, areas or counts of macrophages, extracellular matrix, smooth muscle cell number in the plaque, and sizes of necrotic lesions themselves. This clearly could be done and should have been done using 16-week treated mice. Thus, it would be very important for the authors to show that inhibitory Ab to PDGFD actually can change these findings in culture cells and changes in smooth muscle cell migration and proliferation can make a difference to the pathology of atherosclerosis.*

Regarding in vitro data showing the effectiveness of the Pdgfd blocking antibody, we now show in the revised manuscript that the blocking antibody 25E17 provided by Amgen has a highly significant affect in vitro, blocking migration and proliferation of human aortic smooth muscle cells (HASMC) (see **Reviewer Figs. 5, 6**, and new manuscript **Suppl. Figure 7A, 7B**).

Regarding the lack of lesion characterization with Pdgfd Ab treated mice, we completely agree with the Reviewer, it would have been wonderful to have such data. Unfortunately, we were not provided enough antibody to do an adequately powered lesion histology study. We did attempt a small study with a limited number of mice, but there was considerable attrition in the study groups that severely restricted the number of animals that we could characterize. We ended up with only 2-3 mice in each group, with mortality likely due to the repeated antibody administration. Amgen did not have a large supply of the best 25E17, which was provided to us to help elucidate the mechanism of effect with single cell studies.

Further, it would seem a monumental task to try to identify a commercially available Pdgfd antibody and attempt to conduct the requested in vivo blocking lesion characterization studies. We would need to source multiple commercially available Pdgfd antibodies, investigate them for blocking function, specificity, the dose-response of their blocking effects on smooth muscle migration and proliferation in vitro, as well as their in vivo volume of distribution, pharmacodynamics, etc. These studies should certainly be attempted to benchmark the likely therapeutic potential of such a reagent, but financially prohibitive and beyond the scope of the current work for this academic lab. Our goal for these studies has been to investigate the epigenetic mechanism by which *PDGFD* is associated with risk for coronary artery disease, and to characterize the cellular and molecular mechanisms by which this growth factor can modulate disease pathophysiology in a murine model.

Pages 28-29, Line 748-768:

Pdgfd antibody blockade of SMC proliferation

Human aortic smooth muscle cells (HASMC) were plated at 5000 cells per well in 96-well plates and left to attach and spread for 30 hours. Cells were then washed with serum free media 2 times and incubated in serum free media for 6 hours. PDGF-DD (R&D systems, #1159-SB) was solubilized in 4mM HCl (Vehicle). PDGF-DD (17.85nM) was preincubated for 20 mins with or without anti-PDGF-DD antibody 25E17 (35.7nM) in serum free medium. Cells were treated with PDGF-DD with or without antibody in serum free media. Vehicle was added in control wells. Live cell proliferation was measured with the Incucyte Live-Cell Analysis system (Essen Bioscience). All data is expressed as the mean \pm standard error of the mean (SEM). Statistical analysis performed using a repeated measures one-way ANOVA with Tukey correction.

Pdgfd antibody blockade of SMC migration

Human aortic smooth muscle cells (HASMC) were plated at 8000 cells per well in upper chamber of 96-well transmigration plates from Incucyte (cat # 4648) and left to attach and spread for 30 hours. Cells were then washed with serum free media 2 times and incubated in serum free media for 12 hours. PDGF-DD (R&D systems, #1159-SB) was solubilized in 4mM HCl (Vehicle). PDGF-DD (17.85nM) was preincubated for 20 mins with or without anti-PDGF-DD antibody 25E17 (35.7nM) in serum free medium. PDGF-DD with or without antibody was added to the lower chamber of transmigration plates. Vehicle was added in control

wells. Live cell migration was measured with the Incucyte Live-Cell Analysis system (Essen Bioscience). All data is expressed as the mean \pm standard error of the mean (SEM). Statistical analysis performed using a repeated measures one-way ANOVA with Tukey correction.

Page 15, Line 393-395 Results:

Its blocking activity was validated with in vitro studies with human aortic smooth muscle cells, which showed decreased proliferation and migration in response to PDGFDD in the presence of antibody (Suppl. Figs. 7A, 7B).

2. If the authors cannot show a definitive effect of inhibitory PDGFD and changes in rs2019090 in the ApoE deletion mice, then the findings are very strong for biochemical and molecular analysis of the role of PDGFD and this loci, but they certainly do not strongly support their importance on causality of atherosclerosis, even in mouse model of atherosclerosis. Thus, for the present study, the authors need to remove most, if not all, the references to causality in their statements and provide more pathological data as described above.

We agree with the Reviewer and, as noted above, and have removed all references to causality for disease risk (see redline manuscript). While we cannot provide lesion histology analysis which would further promote investigation of PDGFD blocking reagents as therapeutics for treatment of CAD, we feel that the blocking antibody studies do validate the single cell RNAseq findings in the gene targeted *Pdgfd* mice, and support the identification of likely mechanisms of disease risk. To our knowledge, this is the first single cell study that has employed such validation of gene targeting single cell analysis.

REFERENCES CITED

1. Erdmann J, Kessler T, Munoz Venegas L, Schunkert H. A decade of genome-wide association studies for coronary artery disease: the challenges ahead. *Cardiovasc Res* **114**, 1241-1257 (2018).
2. Klarin D, et al. Genetic analysis in UK Biobank links insulin resistance and transendothelial migration pathways to coronary artery disease. *Nat Genet* **49**, 1392-1397 (2017).
3. Nelson CP, et al. Association analyses based on false discovery rate implicate new loci for coronary artery disease. *Nat Genet* **49**, 1385-1391 (2017).
4. van der Harst P, Verweij N. The Identification of 64 Novel Genetic Loci Provides an Expanded View on the Genetic Architecture of Coronary Artery Disease. *Circ Res* **122**, 433-443 (2017).
5. Gladh H, et al. Mice Lacking Platelet-Derived Growth Factor D Display a Mild Vascular Phenotype. *PLoS One* **11**, e0152276 (2016).

6. Mancini GB, et al. Angiographic disease progression and residual risk of cardiovascular events while on optimal medical therapy: observations from the COURAGE Trial. *Circ Cardiovasc Interv* **4**, 545-552 (2011).
7. Fearon WF. Is a myocardial infarction more likely to result from a mild coronary lesion or an ischemia-producing one? *Circ Cardiovasc Interv* **4**, 539-541 (2011).
8. Little WC, et al. Can coronary angiography predict the site of a subsequent myocardial infarction in patients with mild-to-moderate coronary artery disease? *Circulation* **78**, 1157-1166 (1988).
9. Ridker PM, et al. Antiinflammatory Therapy with Canakinumab for Atherosclerotic Disease. *N Engl J Med* **377**, 1119-1131 (2017).
10. Libby P, Ridker PM, Hansson GK, Leducq Transatlantic Network on A. Inflammation in atherosclerosis: from pathophysiology to practice. *J Am Coll Cardiol* **54**, 2129-2138 (2009).
11. Bennett MR, Sinha S, Owens GK. Vascular Smooth Muscle Cells in Atherosclerosis. *Circ Res* **118**, 692-702 (2016).
12. Abedin M, Tintut Y, Demer LL. Vascular calcification: mechanisms and clinical ramifications. *Arterioscler Thromb Vasc Biol* **24**, 1161-1170 (2004).
13. Hansson GK. Inflammation, atherosclerosis, and coronary artery disease. *N Engl J Med* **352**, 1685-1695 (2005).

Reviewer #1 (Remarks to the Author):

Thank you for responding to my concerns.

Just one thing, as for my question #5. "Fig2A-I, N=3 for each group was small. While you described that you chose three representative ones," could the author show the Reviewer Figs. as a Supplementary Figure?

Because there were three replications, I think you should show them to the reader and convince them.

Reviewer #3 (Remarks to the Author):

The manuscript by Kim et al. has made a good effort to answer reviewer's questions. However, additional effort need to be exerted to answer some simple requests, such as those in Reviewer 1's question 6 regarding empty cassettes.

For Reviewer 3's comment, authors have removed the word 'causality', which is appropriate. In addition, there was a request for providing in vivo data using the inhibitory antibody 2PDGFD in the ApoE^{-/-} mice. The author provided an explanation regarding the lack of sufficient inhibitory antibody VEGF availability. Instead, they provided in vitro studies, which supported the idea, but clearly not direct evidence. Lacking this direct data, it would be very important that the authors state strongly in the Conclusion that the in vitro data using the inhibitory antibody to 2PDGFD are supportive only of the conclusion. Conclusive in vivo evidence will be needed to validate their hypothesis. I don't agree from their last statement that this is the first single cell study that has validated gene targeting single cell analysis, since authors have not validated their findings by in vivo studies as described above.

REVIEWER / EDITOR COMMENTS #2 - NCOMMS-22-35914 - “Molecular mechanisms of coronary artery disease risk at the PDGFD locus”

EDITOR'S COMMENTS

1- We therefore invite you to revise your paper one last time to address the remaining concerns of our reviewers and our editorial requests in the attached documents. Notably, please tone down claims related to antibody data throughout your manuscript in alignment with the request from Reviewer #3.

Thank you for the opportunity to revise and resubmit our revised documents for your consideration. We have edited the text in the manuscript to comply with the concerns of Reviewer #3, as shown here and in the comments to the Reviewer 3. In addition, we have added text to the “limitations” paragraph in the Discussion section of the manuscript to specifically make the points the Reviewer asked to be addressed. This text is shown here and also provided below.

~~Blocking Pdgfd function in the mouse atherosclerosis model validates the molecular and cellular mechanisms of disease risk~~

was changed to:

Page 15, line 379:

“Blocking Pdgfd function in the mouse atherosclerosis model revealed disease related transcriptomic changes similar to those identified in the knockout model”

~~To verify the effects of PDGFD toward disease pathophysiology as identified with the constitutive Pdgfd mouse knockout model...~~

was changed to:

Page 15, line 381:

“To further investigate the transcriptomic effects of PDGFD in the disease setting...”

Page 21, line 534:

Importantly, while the Pdgfd blocking antibody studies provide evidence that the transcriptomic changes associated with knockout are representative of the proposed cellular effects of this gene in the disease setting, they do not provide direct evidence of disease causality. Nonetheless, the *in vitro* effects of the antibody toward SMC function suggest that this approach might be a rational *in vivo* method to prove causality and to investigate novel therapeutic avenues that blocking this pathway may provide. While beyond the scope of this study, further in-depth *in vivo* disease model studies investigating the effects of such blocking antibodies would be required to establish this possibility.

2 - Regarding Reviewer #3's comment upon Reviewer #1's point #6 from the previous round of review, please include non-normalized data in the Supplementary Information file with the empty vector as a separate column along with a reference to this data in the main text. Please include the numbers underlying both the original figure and the supplementary figure in the Source Data file.

The Reviewer #1s point #6 in the previous round of reviews was as follows:

“6. Fig2CD - You performed t-tests in those figures, but I don't know which samples to compare. Moreover, in the siRNA Knockdown experiment and the viral vector transfection experiment, the data should include an empty cassette as a control.”

We felt that it would be simpler to provide the more complete data in the main Figure 2c and 2d. To that end we have replaced the original Figs. 2c and 2d with graphs that show the expression of PDGFD and lncRNA

AP002989 relative to the empty cassette data, which is now shown in the graphs as grey bars and labeled as control (Ctl). In addition, the full dataset for these experiments is provided in the Source Data file. The figure legend has been modified to reflect the updated graphs. Please see the new Fig 2c, 2d panels and legend text embedded below. We very much hope that this is an acceptable solution to addressing the Reviewer concerns.

3 - At the same time we ask that you edit your manuscript to comply with our policies and formatting requirements and to maximise the accessibility and therefore the impact of your work.

We have extensively revised the manuscript to comply with the policies and formatting requirements of your journal, and provided additional experimental details and assurances in the Checklist and the Reporting Summary forms provided.

REVIEWERS' COMMENTS

Reviewer #1 (Remarks to the Author):

Thank you for responding to my concerns.

Just one thing, as for my question #5. "Fig2A-I, N=3 for each group was small. While you described that you chose three representative ones," could the author show the Reviewer Figs. as a Supplementary Figure? Because there were three replications, I think you should show them to the reader and convince them.

Figs. 2a and 2b have now been replaced with graphs that show all biological replicate data combined and the statistical analyses (see edited panels above). These graphs are those previously provided to the Reviewer as a figure in the previous rebuttal document. The combined biological replicate data are now provided in the Data Source File.

Regarding the additional graphs in the panels in Figure 2 (Figs. 2c-2i), they already represented multiple biological replicates, at least three individual experiments combined, with at least three technical replicates per experiment. These are not representative experiments. This distinction is now clear in the Figure 2 legend. All these data are provided in the Source Data file.

To further address the comments of Reviewer #1, for the CRISPRi experiments presented in Figs. 1f and 1g we now include multiple biological replicates, each representing multiple technical replicates (see edits panels reproduced here). The figure legend has been edited as shown here, and these data have been added to the Source Data file.

Reviewer #3 (Remarks to the Author):

The manuscript by Kim et al. has made a good effort to answer reviewer's questions. However, additional effort need to be exerted to answer some simple requests, such as those in Reviewer 1's question 6 regarding empty cassettes.

This comment by Reviewer #1 was related to the absence of a control bar in the graphs representing empty cassette in Fig. 2c and 2d. To address this point, we have revised the Figure 2c and 2d graphs. These graphs now show the expression of *PDGFD* and lncRNA *AP002989*, relative to the empty cassette data, which is now shown in the graphs as the grey bars and labeled as control (Ctl) (see above panels). In addition, the full dataset for these experiments is provided in the Source Data file. The figure legend has been modified to reflect the updated graphs.

For Reviewer 3's comment, authors have removed the word 'causality', which is appropriate. In addition, there was a request for providing in vivo data using the inhibitory antibody 2PDGFD in the ApoE^{-/-} mice. The author provided an explanation regarding the lack of sufficient inhibitory antibody PDGFD availability. Instead, they provided in vitro studies, which supported the idea, but clearly not direct evidence. Lacking this direct data, it would be very important that the authors state strongly in the Conclusion that the in vitro data using the inhibitory antibody to PDGFD are supportive only of the conclusion. Conclusive in vivo evidence will be needed to validate their hypothesis.

~~Blocking Pdgfd function in the mouse atherosclerosis model validates the molecular and cellular mechanisms of disease risk~~

was changed to:

Page 15, line 379:

"Blocking Pdgfd function in the mouse atherosclerosis model revealed disease related transcriptomic changes similar to those identified in the knockout model"

~~To verify the effects of PDGFD toward disease pathophysiology as identified with the constitutive Pdgfd mouse knockout model...~~

was changed to:

Page 15, line 381:

To further investigate the transcriptomic effects of PDGFD in the disease setting...

Fig. 1f, 1g. CRISPRi epigenetic silencing by transduction of dCas9KRAB and single guide RNAs targeted around rs2019090 in cell lot 59386145, a HCASMC primary culture with AA genotype. Expression of *PDGFD*, ** p=0.0033, **** p<0.0001 and lncRNA *AP002989.1* were evaluated by quantitative RT-PCR. Data were normalized relative to control and expressed as mean \pm s.e.m. with p-values obtained using ordinary one-way ANOVA with Dunnett's multiple comparisons post-hoc test. Dots represent three technical replicates from three biologically independent samples. Source data are provided in the Source Data File.

Page 21, line 534:

Importantly, while the Pdgfd blocking antibody studies provide evidence that the transcriptomic changes associated with knockout are representative of the proposed cellular effects of this gene in the disease setting, they do not provide direct evidence of disease causality. Nonetheless, the *in vitro* effects of the antibody toward SMC function suggest that this approach might be a rational *in vivo* method to prove causality and to investigate novel therapeutic avenues that blocking this pathway may provide. While beyond the scope of this study, further in-depth *in vivo* disease model studies investigating the effects of such blocking antibodies would be required to establish this possibility.

I don't agree from their last statement that this is the first single cell study that has validated gene targeting single cell analysis, since authors have not validated their findings by in vivo studies as described above.

This sentence is not included in the revised manuscript, or any related documents.